# Modular deep learning approach for wind farm power forecasting and wake loss prediction

Stijn Ally[1,2], Timothy Verstraeten[1,2], Pieter-Jan Daems[2], Ann Nowé[1], and Jan Helsen[2,3]

[1]Artificial Intelligence Lab Brussels, Vrije Universiteit Brussel, Pleinlaan 9, 1050 Elsene, Belgium
[2]Faculty of Engineering, Vrije Universiteit Brussel, Pleinlaan 2, 1050 Elsene, Belgium
[3]Flanders Make@VUB, Pleinlaan 2, 1050 Elsene, Belgium

**Correspondence:** Stijn Ally (stijn.ally@vub.be)

**Abstract.** Power production of offshore wind farms depends on many parameters and is significantly affected by wake losses. Due to the variability of wind power and its rapidly increasing share in the total energy mix, accurate forecasting of the power production of a wind farm becomes increasingly important. This paper presents a novel data-driven methodology to construct a fast and accurate wind farm power model. The deep learning model is not limited to steady-state situations, but captures also
5  the influence of temporal wind dynamics and the farm power controller on the power production of the wind farm. With a multi-component pipeline, multiple weather forecasts of meteorological forecast providers are incorporated to generate farm power forecasts over multiple time horizons. Furthermore, in conjunction with a data-driven turbine power model, the wind farm model can be used also to predict the wake power losses. The proposed methodology includes a quantification of the prediction uncertainty, which is important for trading and power control applications. A key advantage of the data-driven approach is the
10  high prediction speed compared to physics-based methods, enabling its use in applications that require forecasting multiple scenarios in real-time. It is shown that the accuracy of the proposed power prediction model is better than for some baseline machine learning models. The methodology is demonstrated for two large real-world offshore wind farms located within the Belgian-Dutch wind farm cluster in the North Sea.

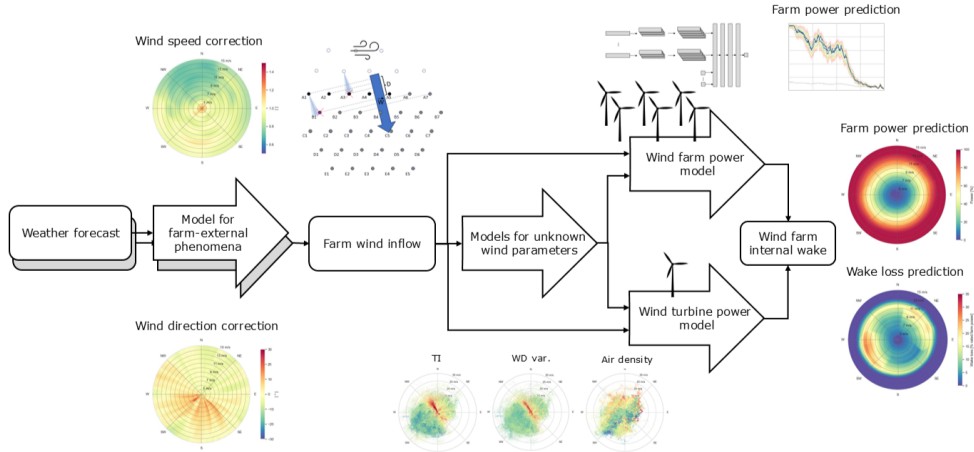

# 1 Introduction

Over the last decades, renewables have been expanding quickly, but the global energy crisis has kicked them into an new phase of even faster growth. Global wind capacity is expected to almost double the upcoming five years, with offshore projects accounting for one-fifth of the growth (IEA, 2022).

Due to the rapidly increasing share of wind power in the total energy mix and its variable and intermittent nature, accurate forecasting of the power production of wind farms becomes increasingly important for wind farm operators, balancing responsible parties (BRP) and transmission system operators (TSO). In addition, as wind turbines and wind farms are getting larger in scale, their influence on the wind flow and the resulting wake effects are becoming more pronounced.

The energy production of a wind farm is influenced by many factors (Lee and Fields, 2020). First of all, the energy production of an individual *wind turbine* depends on many parameters. The type of the turbine (with its specific mechanical and electrical design and control systems) and the wind speed are the most important parameters. Also wind turbulence and air density have an important influence on the power production. Furthermore, a turbine can be derated due to technical reasons or be limited in power by a wind farm power controller.

Secondly, the energy production of a *wind farm* can be significantly reduced by wake losses. Wind turbines modify the air flow for downstream turbines, resulting in velocity deficits and increased turbulence. Reduced velocity of the air results in power losses, whereas increased turbulence results in a faster recovery of the velocity deficit (Sanderse et al., 2011). The magnitude of wake losses depends mainly on the layout of the wind farm (distance between the turbines, orientation of turbine rows and length of the cross-section of the farm) and the wind direction (Barthelmie et al., 2009), but also on other parameters, such as the wind turbulence intensity, atmospheric stability and surface roughness. The power loss of a downstream turbine can reach 40% in full-wake conditions (Barthelmie et al., 2009). When averaged over all different wind directions and wind speeds, annual power losses of a complete farm due to wake can range between 5% to 20%. Studies indicate that active wake control by applying yaw and/or induction control on the turbines may reduce wake losses and hence increase the power production of the farm (Boersma et al., 2017; Verstraeten et al., 2021). However, validation of the possible power gain under real-case conditions in a wind farm is still an active field of research (Bossanyi and Ruisi, 2021; Fleming et al., 2016, 2020).

Thirdly, the characteristics of the *wind inflow* into a wind farm (wind speed, wind direction, turbulence intensity, air density) are influenced by the environment surrounding the wind farm. Upstream wind farms may cause a reduction of the wind speed and an increase of the turbulence intensity (Porté-Agel et al., 2020; Pettas et al., 2021). Wind blockage by neighboring farms and the wind farm itself may also reduce the inflow speed and deflect the inflow direction (Porté-Agel et al., 2020; Bleeg et al., 2018; Strickland et al., 2022). Also the proximity of coast lines can exert a significant influence on the wind characteristics, primarily attributable to differences in roughness and heat capacity between sea and land (Van Der Laan et al., 2017).

## 1.1 Wind and wake flow modeling

Modeling of wind and wake flows within wind farms is challenging and since decades an active field of research. In addition to affecting the power production, wake turbulence also impacts the loading of the turbines (Nejad et al., 2022). A wide range

of different models have been developed. Based on the amount of detail they capture, they can be classified in three classes: low fidelity, medium fidelity and high fidelity models. Each of these types of models have some advantages and drawbacks. Low fidelity models describe only the dominant wake characteristics and are mostly limited to steady state simulations and homogeneous 2D wind inflows. These models are usually relatively fast (order of seconds on personal computer (PC) for a steady state simulation), but need tuning of some hyper-parameters and have a lower accuracy (Jensen Park model (Jensen, 1983), FLOw Redirection and Induction in Steady State (FLORIS) (NREL, b), curled wake model (Martínez-Tossas et al., 2019, 2021) and TurbOPark (Nygaard et al., 2020)). In recent years, some further developments have made it possible to model also some heterogeneous and dynamic environmental conditions (e.g. FLORIDyn (Gebraad and Van Wingerden, 2014; Becker et al., 2022), UFloris (Foloppe et al., 2022)). At the other side of the spectrum, high fidelity models, based on the 3D Navier-Stokes equations, describe flows in high detail. Large-eddy simulations (LES) resolve these equations on a coarse mesh and approximate smaller scale eddies with subgrid models (Sanderse et al., 2011). The main drawback of this type of models is the high computing load (order of days on a computing cluster). Some examples of high fidelity simulators are SOWFA (NREL, c), PALM (University of Hannover) and SP-Wind (Sood et al., 2022). Finally, medium fidelity models (such as DWM (Larsen and Bingoel, 2007), FAST.Farm (NREL, a), WakeFarm (Schepers, 1998)) are based on simplifications of the Navier-Stokes equations. Despite, the simplifications, the computing time for medium fidelity models remains significant (order of minutes on PC) (Boersma et al., 2017).

In contrast to physics-based models, data-driven techniques are not based on prior knowledge about the physical behaviour of the turbines or the air flow. Instead, they focus on fitting a general model on data. Recent developments related to deep learning resulted in a significant leap forward in the modelling of large and complex data sets (LeCun et al., 2015).

*Wind farm operators* have usually huge amounts of historical Supervisory Control and Data Acquisition (SCADA) data acquired by the instrumentation on their turbines. This data can be leveraged by machine learning techniques (Verstraeten et al., 2019). Taking advantage of this large amount of detailed information, accurate models of the wind farm and turbines can be built depending solely on this farm-specific data, and not on any other, often less specific or less accurate, data sources such as theoretical turbine power curves and weather forecast data.

*Commercial weather forecast service providers* (e.g. Stormgeo, KMI) provide nowadays weather forecasts for specific geographical locations. However, forecasts by distinct providers can differ among each other significantly, as they can be based on different weather models and data. In addition, although they provide forecasts for a specific geographical location such as the position of an offshore wind farm, they typically do not take into account the influence of the immediate surroundings of the wind farm, such as the presence of neighboring offshore wind farms and the influence of coast lines. Moreover, they may not be able to provide all weather parameters that are required as inputs for an accurate wind farm power model, and the provided variables may be calibrated differently than the instrumentation data used during the training phase of the data-driven model.

## 1.2  Problem statement

The problem statement that this paper addresses, is to construct a fast and accurate farm power forecasting model, which captures the temporal dynamics of the wind inflow as well as the behaviour of the farm power controller. A deep learning

now-casting power model is trained solely with SCADA data from the wind farm itself. Therefore, this farm power model is unique and independent from any weather forecasting data. In a multi-component pipeline, this single farm power model can be interfaced with multiple weather forecast services (from third-party providers) to forecast the wind farm power over multiple time horizons. The proposed model is capable of forecasting the farm power in a few milliseconds on PC, and this

with an accuracy that surpasses the accuracy of other state-of-the-art data-driven models. This demonstrates that the proposed farm power forecasting model can be used for applications that require both simulation of the farm power controller and fast power forecasting, such as, for example, in a reinforcement control setting where fast evaluations of many possible controller actions are required.

## 1.3 State-of-the-art of data-driven wind power forecasting

This work *distinguishes* itself from other publications about data-driven wind power forecasting on several aspects.

In literature, many models can be found for the power prediction of an individual turbine (Perez-Sanjines et al., 2022; Zehtabiyan-Rezaie et al., 2023; Ti et al., 2021; Kisvari et al., 2021; Lin and Liu, 2020; Daenens et al., 2024) (whether or not superposed afterwards into a farm). The model proposed in this paper predicts the power of a complete wind farm as a whole, without the need for a model for the individual turbines.

Whereas wind farm operators possess typically huge amounts of SCADA data, due to confidentiality reasons, such data is rarely accessible to the research community. Consequently, in many publications about wind farm power and wake modeling, simulation data is generated with physics-based models (such as the low-fidelity static model Floris (Yin and Zhao, 2019; Park and Park, 2019) or mid-fidelity models (Zehtabiyan-Rezaie et al., 2023; Ti et al., 2021)), often only for relatively small virtual wind farms and for a limited set of wind conditions. In contrast, the data used in this paper is SCADA data from two large

real-world offshore wind farms.

Many data-driven models in literature have only a limited set of input features (e.g. restricted to wind speed (Perez-Sanjines et al., 2022) and wind direction or historic wind power series (Wang et al., 2017)), whereas the wind farm power model presented in this paper includes many more input parameters, all having a direct physical influence on the farm power (i.a. air density, turbulence intensity, wind direction variance, wind farm power setpoint and power limitations due to technical

reasons). It is shown that adding these additional input parameters, improves significantly the accuracy of the model.

Data-driven wind farm power models found in literature are often based on data with a relatively coarse time resolution (10 minutes or 1 hour) (Liu et al., 2021; Kisvari et al., 2021; Wang et al., 2021). However, in this work, the wind farm power prediction model is trained with time-series of 1-minute SCADA data in order to be able to capture the temporal dynamics of the inflow wind and the effect of setpoint changes of the farm power controller (for which 1-minute SCADA data was available

for this work). In particular, the influence of inflow wind speed variations on the farm power production is analyzed in this work.

A multi-horizon data-driven wind power forecasting method based on time series forecasting is proposed by Pombo et al. (Pombo et al., 2021). This method results only in good predictions for short time horizon forecasting. In order to obtain long-term multi-horizon forecasts, the multi-component pipeline proposed in this work incorporates weather forecast data for

multiple time horizons as a separate component, without modification of the now-casting farm power model trained solely with SCADA data from the wind farm. Another advantage of the multi-component pipeline, is that the sensitivity to multiple physical input features can be analyzed. This is not possible for a model that consists of one single black box. Sensitivity plots can be interpreted easily by wind energy professionals and increase the interpretability and explainability of the model.

In contrast to most other publications about farm power forecasting, the methodology proposed in this paper, includes also a quantification of the prediction uncertainty. Indeed, not only the accuracy of the model is essential for trading applications and to to ensure reliable operation of wind farms being safety-critical systems, but insight in the uncertainty of the power predictions is crucial as well (Meyers et al., 2022; Braun et al., 2024).

In addition to farm power forecasting, in this work it is demonstrated also how farm-internal and farm-external power losses can be identified based on machine learning (ML) models.

The paper is organized as follows. First, the modular data-driven methodology is described in Section 2. Then, the results of the methodology applied to two offshore wind farms are presented and discussed in Section 3. Finally, the main conclusions of the paper are summarized in Section 4.

## 2 Methodology

The proposed modular data-driven methodology is based on data from multiple data sources and integrates several deep learning models with each other. The used data is described in section §2.1, while the modular structure of the approach is presented in §2.2. The machine learning models incorporated into the modular pipeline are detailed in section §2.3. Section §2.4 explains how the farm internal wake is quantified. Finally, in section §2.5, some baseline models are introduced to benchmark the performance of the wind farm power model proposed in this paper.

### 2.1 Data

The proposed data-driven approach is based on multiple data sources: SCADA data from the wind turbines, from the wind farm power controller and from a weather station located in the wind farm, as well as weather forecasts from multiple weather forecast providers. Typically, such data sources have different time resolutions and accuracy levels. Measurement data from local measurements is typically more accurate than forecast data. Indeed, the accuracy of the former depends solely on the accuracy of the measurement instruments, whereas forecasts data depends on large-scale models and observations with a wide spacial grid that may be wider than a complete wind farm. The time resolution of the data sources used for the showcases in this paper are shown in Table 1.

The optimal time resolution for a wind farm power model depends, on the one hand, on the purpose of the model (e.g. which effects are to be captured by the model) and, on the other hand, may be limited by the available computing hardware

**Table 1.** Data sources

| Data source | Time resolution |
| --- | --- |
| Turbine SCADA data | 1 second |
| Wind farm power controller SCADA data | 1 minute |
| Weather station SCADA data | 10 minutes |
| Weather forecast data from commercial providers | 1 hour |

and required prediction speed. In order to capture the dynamics of a varying wind propagating through the wind farm, the time resolution of the model has to be sufficiently shorter than the duration needed by the wind to cross the entire wind farm. For example, for an offshore wind farm with a cross-section length of 10 km, with wind speeds between turbine cut-in and cut-out wind speeds of respectively 4 m/s and 30 m/s, this duration is between 6 to 42 minutes. So, for that purpose, a sampling time of 1 minute should be adequate. In addition, the farm power set point data available for this work has a 1-minute time resolution, and so setpoint changes of the farm power controller can be captured as well.

In sections §2.1.1 to §2.1.4, each of the data sources used in the proposed methodology is described in more detail.

### 2.1.1 Turbine SCADA data

The turbine SCADA data of the wind farms that are used as showcases in this paper, has a sampling time of 1 second. For both farms, the data of the turbines comprise the following measurements:

– wind speed (measured by an anemometer located on top of the turbine nacelle),

– wind direction (measured by a wind vane located on top of the turbine nacelle), and

– turbine active power (measured at the power terminals of the turbine).

For one of the two farms, an additional data field is available which expresses the maximum power that the turbine could technically produce at the moment in case of sufficient wind:

– turbine active power capability.

The maximum power can indeed be limited below the rated turbine power due to a technical problem of the turbine or by a curtailment imposed from externally (e.g. by the farm power controller).

Based on the wind speed and direction measurement data, two additional data features are built that can be used as measures for the wind turbulence: wind turbulence intensity $\tau$ and wind direction variance $\phi$, as:

$$\tau = \frac{\sigma(v)}{\mu(v)} \tag{1}$$

$$\phi = \sigma^2(\theta_{lat}) + \sigma^2(\theta_{lon}) \tag{2}$$

with $\sigma(v)$ the standard deviation of the wind speed, $\mu(v)$ the average wind speed, and $\sigma^2(\theta_{lat})$ and $\sigma^2(\theta_{lon})$ the variance of respectively the lateral and longitudinal component of the wind direction, each during the 10-minute time interval centered around the 1-second data point.

As the wind farm and the turbine power models proposed in this paper have a temporal resolution of 1 minute, each of the above 1-second data feature sequences is averaged to 1-minute data blocks.

### 2.1.2 Wind farm data

Most of the farm data can be calculated by aggregating the SCADA data from the individual turbines. The active power produced by the farm is calculated as the sum of the individual turbine active powers. Similarly, the farm active power capability is calculated by summing up the power capability of each of the turbines.

Simply adding or averaging the wind measurement data of all individual turbines, however, would lead to a loss of information about the spatial variation of these features throughout the farm. Instead, the characteristics of the wind flowing into the wind farm are used, measured by a subset of turbines located in the upstream part of the farm. This allows also to isolate farm-internal wake and farm-external wake in separate ML models (the latter depending also on the operational status of the neighboring farms). The set of upstream turbines is determined by the following *corrected geographic sorting* algorithm (see Figure 1):

1. calculate wind vector $\overrightarrow{W}$ as average of the wind speeds and directions measured by all the turbines of the farm,

2. based on the farm layout, determine the projection of the position of each turbine on $\overrightarrow{W}$,

3. select all turbines that are located in the most upstream zone of the farm with length $D$ along $\overrightarrow{W}$,

4. if the number of selected turbines is smaller than minimum quantity $N$, complete the set of turbines by adding additional turbines in order of their projected position on $\overrightarrow{W}$,

5. from the set of $N$ turbines, select the subset of $M$ turbines with the highest wind speeds.

As rule-of-thumb, an appropriate choice for parameter $D$, is a value slightly smaller than the distance between the outer turbine rows of the farm. As a consequence, in case that the wind vector $\overrightarrow{W}$ is nearly perpendicular to a side of the wind farm, all turbines in the upstream row will be selected as upstream turbines, enabling to capture possible differences in wind characteristics over the complete width of the farm. In case that the wind vector $\overrightarrow{W}$ is oriented towards a corner of the farm layout, however, only one single or very few turbines may be selected. Therefore, a minimum quantity of $N$ of turbines is selected, with $N$ a value which should be larger than 1 and smaller than the quantity of turbines in a single row. Finally, in order to remove turbines that risk to be located in a narrow waked position caused by a turbine from the farm itself or from a neighboring farm, only the $M$ turbines with the highest wind speeds are retained. The farm inflow wind parameters (inflow wind speed, inflow wind direction, inflow turbulence intensity and inflow wind direction variance) are then calculated as averages of the corresponding features from the selected individual upstream wind turbines. The overall methodology presented

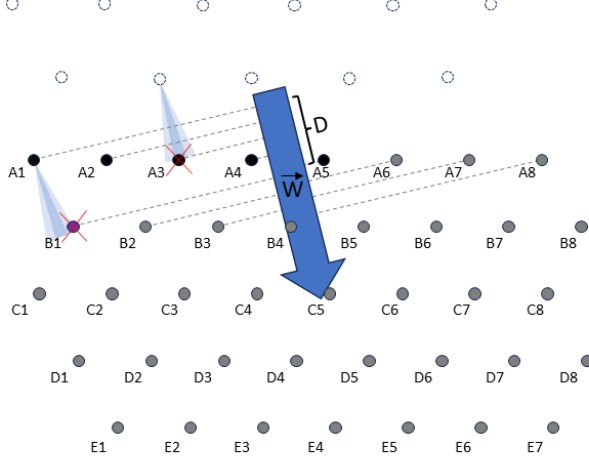

**Figure 1.** Determination of the upstream turbines of the wind farm with turbines A1 to E7, for the case of an average wind $\overrightarrow{W}$ from north-northwest direction. Turbines A1 to A5 are located in the most upstream zone of the farm with length $D$ along $\overrightarrow{W}$. As in this example the minimal quantity of upstream turbines to be taken into account ($N$) equals 6, turbine B1 is added to the set of upstream turbines. Finally, as in this example the quantity of upstream turbines with the highest wind speed to be taken into account ($M$) equals 4, the two turbines with the lowest wind speed are not taken into account for determining the inflow wind characteristics. In this example, these are turbines A3 (which appears to be in a fully-waked position from a turbine of a neighboring wind farm) and B1 (which is in a partially-waked position from turbine A1).

in this paper can also be applied with alternative algorithms for determining the upstream turbines. There is, however, always a trade-off between the responsiveness of the model on wind transients and the average accuracy of the model, due to the spatial variation of the wind characteristics across the wind farm.

Note that the farm inflow wind characteristics, as determined by above algorithm, are independent from the operational status of the wind farm and, in particular, from the control actions of the farm power controller. This is not the case for wind characteristics measured by turbines located in waked conditions more downstream in the wind farm. This independency makes it possible to map weather forecasts to the farm inflow wind characteristics with models that are independent from the wind farm operational status.

An additional farm parameter, having a major influence on the farm power production, is the set-point of the farm power controller. Currently, for the majority of wind farms, active farm power control is still rarely used. Some farms, however, do nowadays already use actively farm power control in order to perform power balancing (Kölle et al., 2022). Due to the fast growing share of wind energy in the energy mix, the use of farm power control may become more predominant in the near future. For the wind farms used as examples in this paper, the farm power set point data has a 1-minute time resolution.

Finally, one additional wind farm feature is composed from the turbine SCADA data: the quantity of stopped turbines. If one or more turbines are stopped (e.g. for maintenance reasons), the total power of the farm is reduced. On the other hand, a

turbine in standstill will not cause wake for other downstream turbines. Depending on the available turbine SCADA data of the farms, the feature "quantity of stopped turbines" is built by counting the number of turbines that is producing zero power (or even consuming power) or as the number of turbines with power capability equal to zero. Notice that, based on this feature, the farm power model does not get the information which turbine(s) specifically are at standstill, which may lead to some loss of accuracy of the model. Indeed, for example, stopping a turbine in a waked position may lead to less power reduction than stopping a turbine in an upstream position with free wind inflow.

### 2.1.3 SCADA weather data

For the showcases in this paper, measurement data from a weather station located in one of the wind farms is used. The data set comprises the air temperature, humidity and pressure. Based on these three measurements, the relative air density is calculated. For the farms used as examples in this paper, the available measurements are data averaged over 10-minute intervals. In order to use the air density as input for the 1-minute farm and turbine power models, the data series is interpolated to a 1-minute data sequence.

### 2.1.4 Weather forecast data

For the showcases in this paper, wind speed and direction forecasts have been used for the location of the wind farms and for a height of 100m (similar to the hub height of the turbines). The forecasts are provided by a commercial weather forecast provider and have a 1-hour time resolution. Based on the lead time of the forecasts, separate data sets have been composed with intra-day forecasts, day-ahead forecasts (before 11 a.m. the day before energy production) and three-days-ahead forecasts.

## 2.2 Multi-component pipeline

Figure 2 shows the overall modular structure of the methodology proposed in this paper, integrating the components described in sections §2.3 and §2.4. Weather forecasts of weather forecast providers are processed to forecasts of the wind inflow experienced by the farm. In doing so, phenomena such as external wake, wind farm blockage, coastal effects and other unknown systematic forecasting errors are accounted for. Based on these corrected inflow wind speeds and directions, possibly unknown wind parameters are estimated using auxiliary models. With all the resulting input parameters, the wind farm power is forecasted. Additionally, the power of an individual turbine operating under identical environmental conditions is forecasted. By subtracting the power forecast of the wind farm from the power forecast of the individual turbine multiplied with the number of turbines in the farm, the wake loss within the wind farm can be quantified.

## 2.3 Machine learning models

In this section, all ML models are described which are part of the modular deep learning pipeline (§2.2). The wind farm power model, which is the core model of this work, is detailed in §2.3.1. This model predicts the real-time wind farm power production based on a set of wind inflow measurements and other farm-specific parameters. The ML model for predicting the

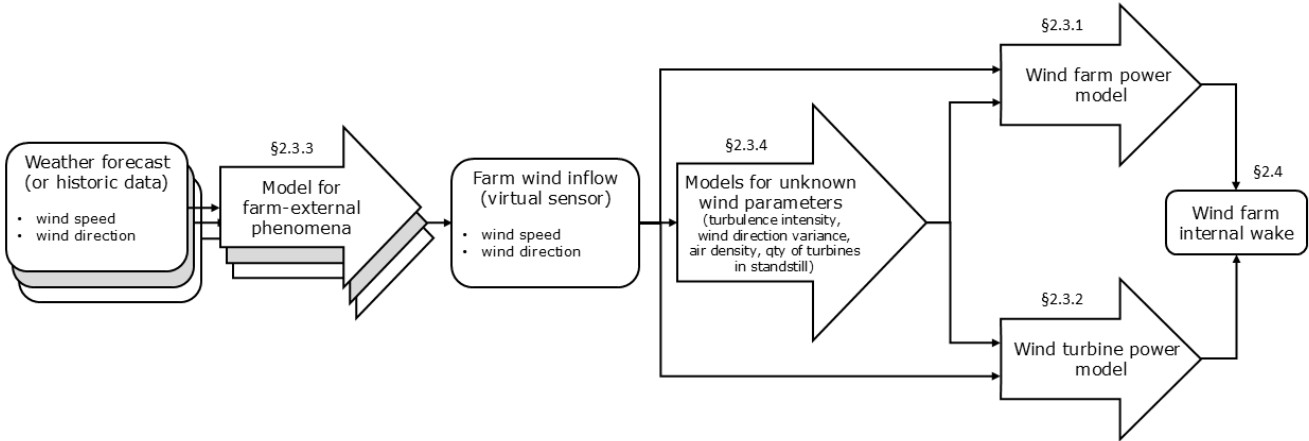

**Figure 2.** Overview of the modular structure of the multi-component pipeline with multiple ML models

real-time power of an individual wind turbine is presented in §2.3.2. The models converting weather forecasts into accurate
forecasts of the wind inflow conditions experienced by the wind farm are detailed in §2.3.3. Lastly, the auxiliary models used
to estimate missing input parameters for the wind farm and turbine power models are discussed in §2.3.4.

### 2.3.1  Wind farm power ML model

The core ML model proposed in this paper predicts the wind farm power based solely on measurement data from conventional
turbine instrumentation. This type of data is usually available to any wind farm operator. Each of the input features of the model
has a direct (physical) influence on the power production of the wind farm.

As it takes typically several minutes up to half an hour for the wind to cross a complete wind farm (depending on the wind
speed and size of the farm), the wind and wake characteristics at time $t_i$ across the farm do not only depend on the wind inflow
at time $t_i$, but also on the evolution of the wind inflow between $t_i$ and $t_i - T$ (with $T$ = 30 minutes). Also control actions
from the farm power controller and the quantity of stopped turbines in the past may have an influence on the spatial wind and
wake profile across the farm. In contrast, some input parameters have an immediate discontinuous impact on the farm power
production, independent of their historical values. For example, a farm power controller can reduce quasi-instantaneously
(i.e. with a response time smaller than or similar to one 1-minute time step) the power of all turbines across the wind farm.
Therefore, the proposed model is composed of two separate parts. For each input parameter that influences the wind and wake
profile across the farm in a continuous way, a sequence of historic values is passed to a convolution branch. The outputs of
these convolution branches are then passed to a feed-forward neural network, together with the input parameters that have
(only) an immediate impact on the wind farm power production.

The structure of the proposed farm power model is shown in Figure 3. $K$ separate convolution branches consist of two 1D
convolutional layers. The outputs of these $K$ branches are passed, together with $L$ additional input features, through a flattening

and dropout layer, to a feed-forward regression neural network composed of three dense layers, each followed by a dropout layer. Each convolution layer has 8 convolution filters ($c_1 = c_2 = 8$) with a kernel size of 3, a stride of 1 and no padding. The fully connected dense layers have respectively 128 ($n1$), 256 ($n2$) and 256 ($n3$) units with a rectified linear (ReLU) activation function.

A *feed-forward neural network* is one of the simplest types of artificial neural networks, where the information flow is in one direction, from the input layer, passing through the hidden layers to the output layer. A *dense layer* is a fully connected layer where every neuron in the layer is connected to every neuron in the previous layer. A *convolution layer* involves the sliding of filters over the input data. A filter is a small matrix with learnable parameters. In the proposed model, each filter, with dimension $1 \times 3$ (= kernel size), slides with a step of 1 (= stride step) over the input time sequences and can detect specific patterns in these sequences that are significant for the farm power production. A *flattening layer* is a layer which transforms a multi-dimensional array into a one-dimensional array. A *dropout layer* is a layer in which randomly a fraction of the neurons outputs are set to zero during training, and so effectively dropping neurons out of the network. This prevents the network from becoming overly reliant on specific neurons, which reduces overfitting and improves the robustness of the model. A *rectified linear activation function* is one of the most widely used activation functions in neural networks, which introduces non-linearity in the network to allow the model to learn complex patterns.

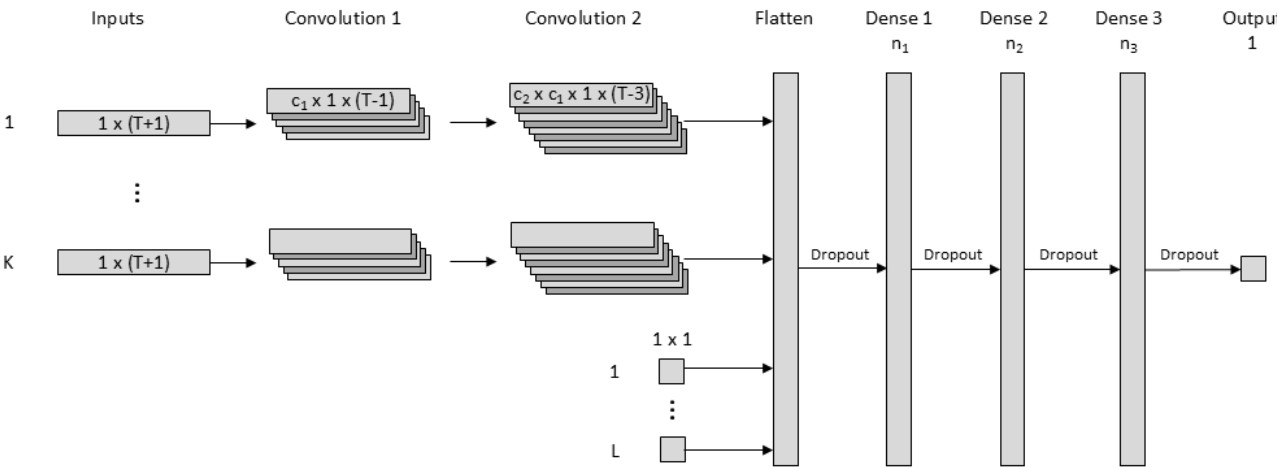

**Figure 3.** Structure of the proposed deep learning farm power model. $K$ separate convolution branches consist of two 1D convolutional layers. The outputs of these $K$ branches are passed, together with $L$ additional input features, through a flattening and dropout layer, to a three-layer regression model that is composed of three dense layers, each followed by a dropout layer.

Time sequences from $t_i$ to $t_{i-T}$ of the following $K$ input parameters (based on SCADA data) are passed to the $K$ convolution branches:

- farm inflow wind speed,

- lateral component of the farm inflow wind direction,

- longitudinal component of the farm inflow wind direction,

- farm inflow turbulence intensity (Equation 1),

- farm inflow wind direction variance (Equation 2),

- air density,

- set point of the farm power controller (if available in the data set),

- quantity of stopped turbines, and

- active power capability of the farm (if available in the data set).

In parallel, the values at $t_i$ of the following $L$ input parameters are passed directly to the feed-forward component of the neural network:

- set point of the farm controller (if available in the data set),

- quantity of stopped turbines, and

- active power capability of the farm (if available in the data set).

Notice that the lateral and longitudinal components of the wind direction are used as inputs, instead of the wind direction itself (expressed in degrees). This is done to guarantee continuity in the data for wind direction 360°/0°.

The dropout layers in the neural network model allow to predict not only the expected power production of the wind farm, but also to quantify the uncertainty of that prediction. The method applied in this paper is referred to as *Monte Carlo dropout* (Gal and Ghahramani, 2016). This is an epistemic method as it quantifies the uncertainty arising from the model architecture 300 and the amount of data.

Instead of generating one single prediction of the farm power for time step $t_i$, $N'$ different power predictions $\hat{P}_i^n$ are generated by using the model with active dropout layers (in the same way as during the training phase of the model). The power prediction $\hat{P}_i$ is then calculated as the average of these $N'$ different power predictions. In addition, also the variance of these $N'$ power forecasts $\sigma_{\hat{P}_i}^2$ can be determined, as well as an arbitrary set of percentiles and thus confidence intervals 305 $[\hat{P}_i - \alpha_i, \hat{P}_i + \beta_i]$.

Unfortunately, the Monte Carlo dropout method is prone to miscalibration, i.e. the predictive uncertainty does not correspond well to the model error. Therefore, a method referred to as *sigma-scaling* is applied, calibrating jointly the epistemic uncertainty from the model and the aleatoric uncertainty from the data (e.g. due to sensor noise) (Laves et al., 2020). For each time step $t_i$ of the test data set, the following ratio is calculated:

$$q_i^2 \quad = \frac{(\hat{P}_i - P_i)^2}{\sigma_{\hat{P}_i}^2} \tag{3}$$

with $P_i$ the true farm power at time step $t_i$. According to Equation 3, $q_i^2$ is thus the ratio of the prediction squared error of the model for time step $t_i$ and the variance calculated with the Monte Carlo dropout method for that time step. Analysis of the results for the wind farms used as example in this paper, shows that the prediction errors $\hat{P}_i - P_i$ and ratios $q_i^2$ depend mainly on the wind speed and the set point of the farm power controller. This could be expected, as for low wind speeds and for high wind speeds above the rated turbine wind speed, the power curve of the turbines is relatively flat. In contrast, for wind speeds slightly below the rated wind speed, the power curve of the turbines is the steepest. Therefore, a calibration function $q^2(v, s)$ is established, with $v$ the wind speed and $s$ the set point of the farm power controller. This is done by mapping a simple feed-forward neural network (with two hidden dense layers with 256 units) to the complete test data set. With this calibration function, the variance predicted with the Monto Carlo dropout method $\sigma^2_{\hat{P}_i}$ is re-calibrated as:

$$\hat{\sigma}^2_{\hat{P}_i} \quad = \sigma^2_{\hat{P}_i} \times q^2(v_i, s_i) \tag{4}$$

Similarly, the confidence intervals generated with the Monte Carlo dropout method are re-scaled as:

$$[\hat{P}_i - \hat{\alpha}_i, \hat{P}_i + \hat{\beta}_i] = \quad [\hat{P}_i - \alpha_i \times q(v_i, s_i), \hat{P}_i + \beta_i \times q(v_i, s_i)] \tag{5}$$

### 2.3.2 Turbine ML model

Based on the same data set as used for the farm power model, also a turbine power model is built. This turbine model has thus also a 1-minute time resolution. However, in order to model a healthy turbine without any technical deration, reduced power mode or curtailment by its power controller, all data points with a reduced power capability and/or curtailment are removed from the training data set.

In contrast to a wind farm, the power production of a single turbine does not depend on the wind speed from multiple antecedent 1-minute time steps. Indeed, the response time of a single wind turbine, which is determined predominately by the inertia of its rotor, is significantly faster. Furthermore, the wind direction has no direct influence on the power of a turbine, as long as the yaw control system orientates the turbine perpendicular to the incoming wind direction. (For the wind farms used as examples in this paper no wake steering is done by applying yaw control).

As input data features, wind speed, turbulence intensity (Equation 1), wind direction variance (Equation 2) and air density are used. The turbine power is modeled by a simple feed-forward neural network, composed of three fully connected dense layers, each with 128 units with a rectified linear activation function.

### 2.3.3 ML models for mapping weather forecasts

Commercial weather forecast services provide nowadays weather forecasts for specific geographical locations, such as the position of a wind farm. Forecasts of different providers can differ due to the use of different weather models or data. Usually, wind speed and wind direction forecasts do not take into account the presence of neighboring wind farms or coast lines. Furthermore, the forecasts may not have exactly the same calibration as the instrumentation that has been used to train the wind farm power model.

Therefore, the weather forecasts should be mapped to the real weather conditions experienced by the wind farm. The appropriate data-driven models for such mappings will depend on the available data sources from weather forecast providers. For the showcases in this paper, the wind speed forecasts are mapped to the measured farm inflow wind speeds, and the wind direction
forecasts are mapped to the measured farm inflow wind directions.

The structure of the ML model that is used for the wind speed mapping is shown in Figure 4 (left). It is a feed-forward regression neural network with three hidden dense layers, each followed by a dropout layer. The dense layers consist respectively of 16 ($n_1$), 32 ($n_2$) and 32 ($n_3$) units with a ReLU activation function. As the time step of the weather forecasts available for this work are rather course (1 hour time step), in a pre-processing step, the 1-hour wind speed forecast data is first interpolated
to wind speed forecast data $\tilde{v}_t$ with a 10-minute time step . In order to provide the mapping model the possibility to capture the possible influence of wind speed variations, not only the wind speed forecast of the specific 10-minute time step $\tilde{v}_t$ is used as input feature, but also the wind speed forecasts of the two preceding and the two following 10-minute times steps ($\tilde{v}_{t-2}, \tilde{v}_{t-1}, \tilde{v}_{t+1}, \tilde{v}_{t+2}$). As output feature, the average inflow wind speed $v_t$ for the 10-minute time step is used.

The structure of the ML model used for the mapping of the wind direction forecasts is the same as for the wind speed
forecasts. The only difference are the input and output features, which are the cosine and sine of the forecasted and measured wind directions, resp. $\cos \hat{\theta}_t$, $\sin \hat{\theta}_t$, $\cos \theta_t$ and $\sin \theta_t$. The structure of the ML model is shown in Figure 4 (right).

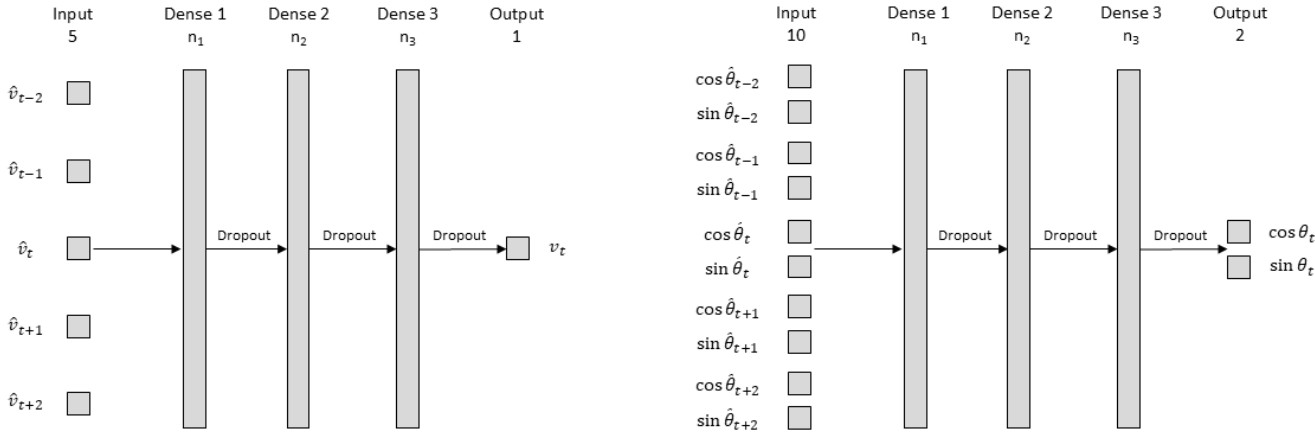

**Figure 4.** Structure of the deep learning models used to map wind speed forecasts to measured inflow wind speeds (left) and wind direction forecasts to measured inflow wind directions (right).

### 2.3.4    Auxiliary ML models for missing input parameters

The farm power model (as specified in §2.3.1) has many input parameters related to the inflow wind. Sometimes the value of some of these wind characteristics is not known, or at least not accurately. For example, weather forecast providers usually do
not provide accurate information about the wind turbulence taking also the presence of neighboring wind farms into account.

In such case, one could decide to use a single average value for these parameters. However, the turbulence intensity, wind direction variance, air density and the number of stopped turbines depend all more or less on the wind speed and wind direction. Therefore, for each of these four parameters a model is built to predict its value based on the wind speed and wind direction. These auxiliary models may be specific for a particular wind farm and are not the focus of this paper.

## 2.4 Farm internal wake loss

The proposed modular approach allows also to predict the power losses in the wind farm due to internal wake (ref. Figure 2). The power loss in a wind farm with $J$ wind turbines due to internal wake, can be calculated as:

$$P_{wake} = \sum_{j=1}^{J} P_{WT}{}^{j} - P \tag{6}$$

with $P_{WT}{}^{j}$ the power production of wind turbine $j$ subjected to wind with the same characteristics as the inflow wind of the farm, and $P$ the farm power. In case of $J$ identical turbines, the farm power loss due to wake can be simplified as:

$$P_{wake} = J \times P_{WT} - P, \tag{7}$$

where $P_{WT} = P_{WT}{}^{j}, \forall j.$ \hfill (8)

Consequently, the wake loss can be modeled using the wind farm power model (as proposed in §2.3.1) and the model for a single turbine (§2.3.2):

$$P_{wake}(\boldsymbol{v}, \boldsymbol{\theta}, \boldsymbol{\tau}, \boldsymbol{\phi}, \boldsymbol{\rho}) = J \times P_{WT}(v, \tau, \phi, \rho) - P(\boldsymbol{v}, \boldsymbol{\theta}, \boldsymbol{\tau}, \boldsymbol{\phi}, \boldsymbol{\rho}) \tag{9}$$

with $v$, $\theta$, $\tau$, $\phi$ and $\rho$ respectively the farm inflow wind speed, wind direction, turbulence intensity, wind direction variance and air density. Notice that for the farm power model, and thus also for the wake model, these wind parameters are 30-minute time sequences.

## 2.5 Baseline models

In order to demonstrate the influence of some key implementation choices of the wind farm power model (§2.3.1), its performance metrics are compared with those of two baseline models that do not comprise these implementation choices.

The first baseline model is a feed-forward neural network model with three hidden dense layers, which is a sub-component of the proposed wind farm power model. This baseline model, however, does not comprise any convolution layer and takes only the wind speed and the sine and cosine of the wind direction at time step $t_0$ as input features. Consequently, this model has no information about the preceding 30 minutes and, by consequence, no information about the variability of the wind parameters. Furthermore, this model has no knowledge about the other measurement data, such as the air density, turbulence intensity and wind direction variation, nor about the farm power controller and the quantity of turbines at standstill.

The second baseline model is identical to the proposed wind farm power model, with the exception that it does not comprise any convolution layer. This baseline model has the same set of input features as the wind farm power model, but, similar to the other baseline model, it does not have information about the variability of the wind parameters.

## 3 Results

The methodology as described in section 2, has been applied to two offshore wind farms located in the Belgian-Dutch wind farm cluster in the North Sea (see Figure 5). This cluster comprises 13 wind farms. It is located on a distance of 20 km to 60 km from the Belgian and Dutch coastline. For confidentiality reasons, the two wind farms are denoted in this paper as *Wind Farm 1* and *Wind Farm 2*. The two farms are operated by different farm operators and have a different type of wind turbines. The main characteristics of the two farms and the complete wind farm cluster are listed in Table 2. All presented results involving farm power and turbine power are normalized on respectively the installed capacity of the wind farm and the rated turbine power.

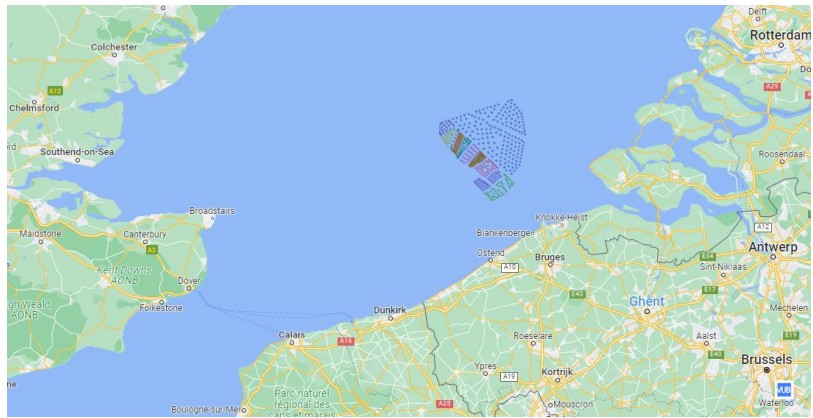

**Figure 5.** Position of the Belgian-Dutch offshore wind farm cluster in the North Sea (© Google Maps)

**Table 2.** Main characteristics of the Belgian-Dutch offshore wind farm cluster and the two wind farms used as examples in this paper.

|  | Quantity of turbines | Installed capacity [MW] | Acreage [km$^2$] |
|---|---|---|---|
| Belgian-Dutch wind farm cluster | 572 | 3764 | 608 |
| Wind Farm 1 | $\geq$40 | $\geq$300 | $\geq$40 |
| Wind Farm 2 | $\geq$30 | $\geq$200 | $\geq$20 |

The results are grouped in the following three sections. In Section 3.1, results are presented related to the core wind farm power prediction model of each of the two wind farms. Section 3.2 focuses on the estimation of farm-internal wake. Finally, in Section 3.3, some results are shown related to the integration of weather forecasts.

### 3.1 Wind farm power model

#### 3.1.1 Data

For both Wind Farm 1 and Wind Farm 2, weather and turbine data was available for a period of about 2.5 years. Table A2 shows the quantity of 1-minute data points resulting from the pre-processing of the SCADA data, and split in a distinct training, test and validation data set.

For splitting the available data in three independent data sets, the following steps have been applied. First, a long time-sequence of consecutive days is selected as validation data set. Thereafter, in order to guarantee that the test and training data 410 sets are sufficiently independent, and, at the same time, are both representative for all seasons, hours-of-the-day and days-of-the-week, the test data set is established by selecting all data from some days-of-the-month from the remaining data. More specifically, the test data set comprises the data from the days 2, 3, 4, 16, 17, 18 and 28 of each month.

**Table 3.** Quantity of data points used for training, test and validation of the two wind farm power models

| Data set | Wind Farm 1 | Wind Farm 2 |
|---|---|---|
| Training | 960695 | 978008 |
| Test | 292996 | 292078 |
| Validation | 15810 | 15810 |

Figures 6 and 7 show the wind roses for the inflow wind of Wind Farm 1 and Wind Farm 2 respectively. South-southeast is 415 the predominant wind direction with also the highest wind speeds. Although the two wind farms are located close to each other in the same wind farm cluster, the wind conditions experienced by the two wind farms are different.

The data plots in figures 8, 10 and 12 illustrate the dependency of the farm power on respectively the wind turbulence intensity, wind direction variance and air density. Higher wind turbulence results in higher farm power, as higher turbulence facilitates the wake recovery reducing so the possible power loss for downstream turbines. A higher air density results also in 420 an increase of the turbine power, because the mass, and thus kinetic energy, of the moving air is higher.

The polar data plots in figures 9, 11 and 13 give an indication of the correlation between respectively turbulence intensity, wind direction variance and air density with the inflow wind speed and wind direction. In the northwest and southeast wind directions, Wind Farm 2 has in its immediate vicinity neighboring wind farms with densely positioned turbines, resulting in a high turbulence intensity for these wind directions. Also in the northeast direction, Wind Farm 2 has a neighbouring wind farm. 425 However, that wind farm is located further away and its turbines are positioned less densely. It can be seen also, that the wind turbulence is lower for high wind speeds than for slow speeds. Furthermore, it can be seen that the southwest direction is the wind direction with the highest wind speeds. In that direction, the wind is coming from over sea in parallel with the coastline, through the narrow Strait of Dover, which has high cliffs. It can be noticed also that wind in parallel to the coast line typically

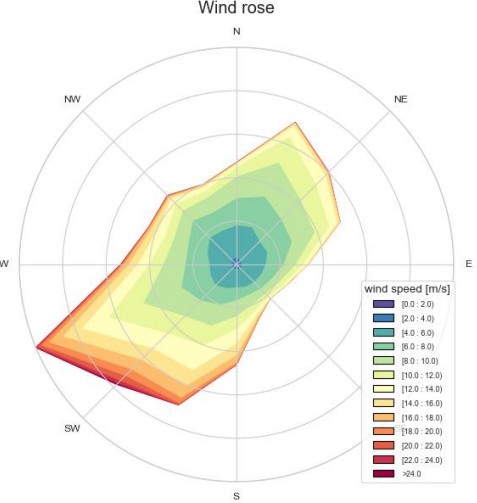

**Figure 6.** Wind rose for Wind Farm 1

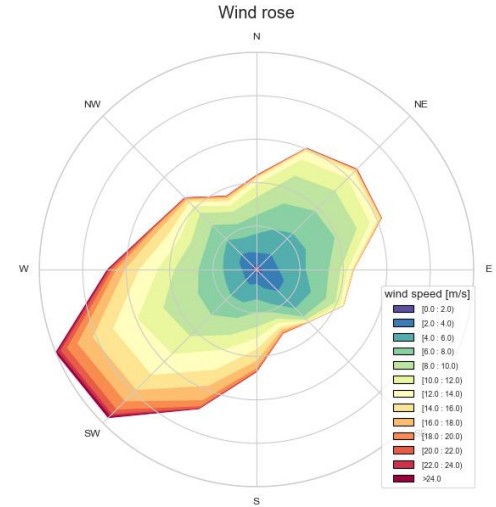

**Figure 7.** Wind rose for Wind Farm 2

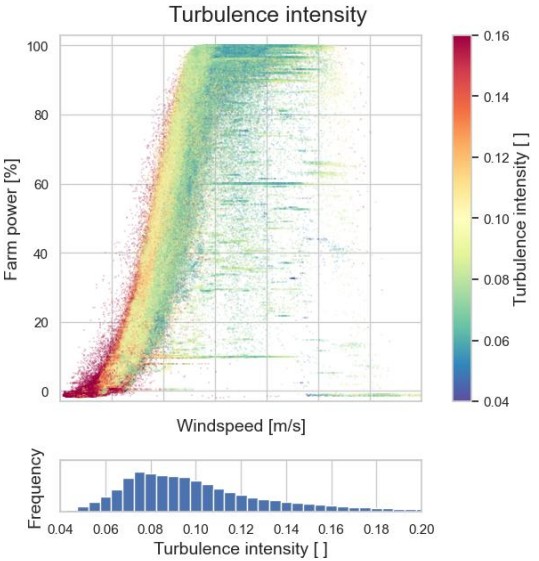

**Figure 8.** Farm power as a function of inflow wind speed and wind turbulence intensity, and frequency distribution of turbulence intensity (Wind Farm 2)

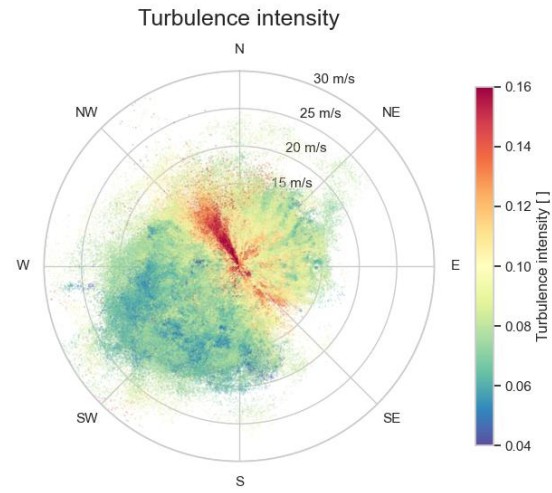

**Figure 9.** Inflow turbulence intensity as a function of wind speed and wind direction (Wind Farm 2)

has a lower air density compared to the orthogonal directions from and to the mainland. This corresponds to the fact that humid air has a lower air density than dry air.

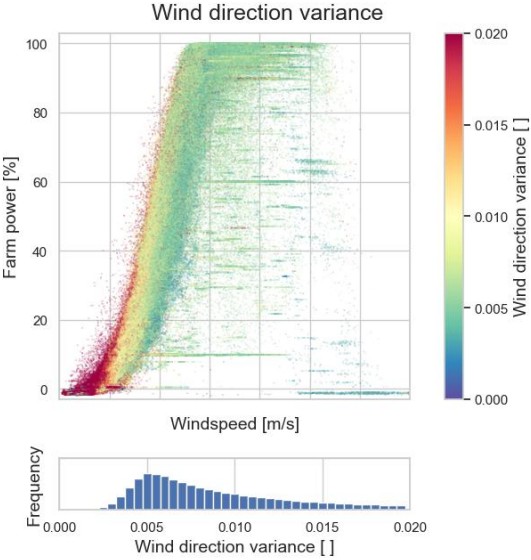

**Figure 10.** Inflow wind direction variance as a function of wind speed and wind direction, and frequency distribution of wind direction variance (Wind Farm 2)

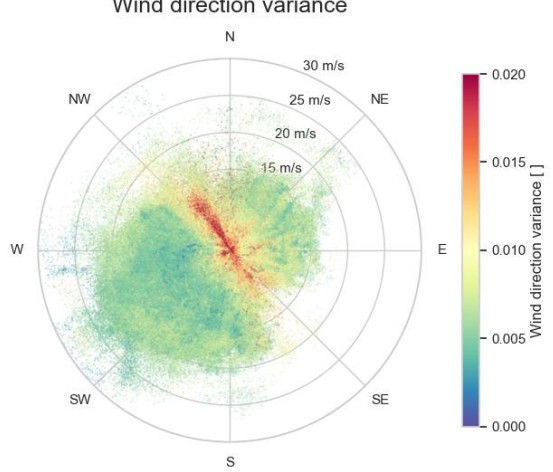

**Figure 11.** Inflow wind direction variance as a function of wind speed and wind direction (Wind Farm 2)

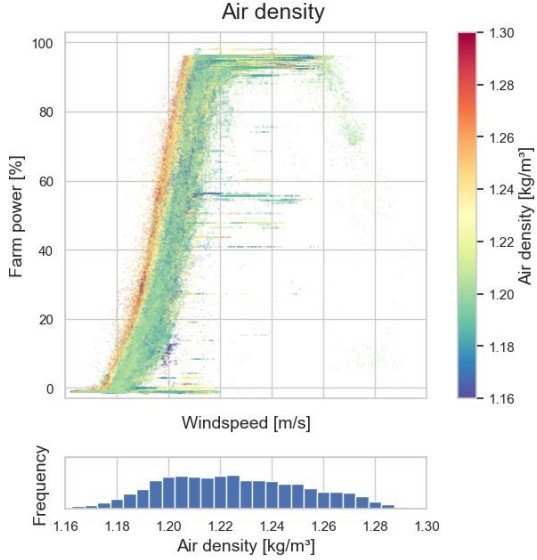

**Figure 12.** Farm power as a function of inflow wind speed and air density, and frequency distribution of air density (Wind Farm 1)

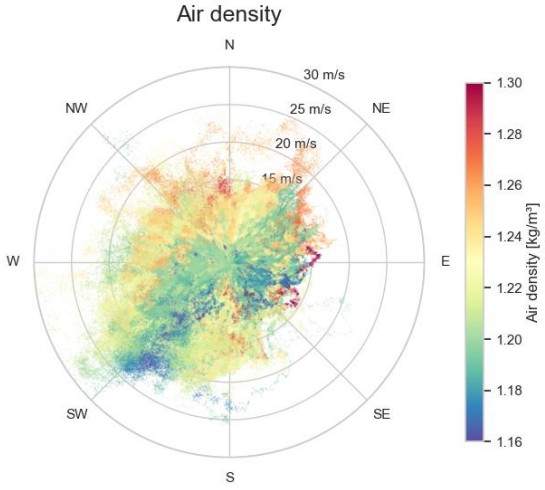

**Figure 13.** Air density as a function of wind speed and wind direction (Wind Farm 1)

When comparing figures 8 and 12 from respectively Wind Farm 2 and Wind Farm 1, it can be seen that Wind Farm 2 has markedly more data points with a reduced power. This is due to the power controller of the wind farm which, in some circumstances, regulates the farm power in a continuous manner. In contrast, for Wind Farm 1, there appear to be farm curtailments to a few specific discrete power set points (such as ∼60% of the maximum farm power). Furthermore, it can be noticed also, that for Wind Farm 1, the total installed turbine capacity is never reached. This is due to the farm power controller, which limits the maximum producible power of the farm.

### 3.1.2 Hyper-parameters

Table 4 shows the parameters used to determine the set of upstream turbines according to the algorithm described in §2.1.2. Wind farm 1 comprises more turbines than Wind Farm 2 (ref. parameter $N$) and these are positioned more widely apart (ref. parameter $D$).

**Table 4.** Parameters used for the selection of the upstream turbines

| Parameter | Wind Farm 1 | Wind Farm 2 |
| --- | --- | --- |
| D | 1000m | 500m |
| N | 12 | 6 |
| M | 10 | 4 |

The hyper-parameters of the wind farm power ML model (§2.3.1) and of the wind forecast mapping ML models (§2.3.3) are documented in Appendix A.

### 3.1.3 Performance metrics

In Table 5, some performance metrics (Piotrowski et al., 2022) of the farm power models for Wind Farm 1 and Wind Farm 2 are listed: the root mean square error normalized on the installed power capacity of the wind farm (nRMSE), the normalized mean absolute error (nMAE), normalized mean error (nME) and the R2-score, for both the training and test data set. The nME for each of the farms is near to 0% and the nMAE is 2.42% and 2.14% for the test data sets of Wind Farm 1 and Wind Farm 2 respectively. The differences between the performance metrics of the training and test data sets are very small, which is a sign for a satisfactory fit (little overfitting).

In Table 6, as a comparison basis, the corresponding performance metrics are shown for the two baseline models described in §2.5. *FNN($v_0, \theta_0$)* denotes the first baseline model, which has only the inflow wind speed $v_0$ and the sine and cosine of the inflow wind direction $\theta_0$ at time step $t_0$ as input features. *FNN(all input features at $t_0$)* denotes the second baseline model,

**Table 5.** Performance metrics of the power models for Wind Farm 1 and Wind Farm 2

|  | Wind Farm 1 | | Wind Farm 2 | |
|---|---|---|---|---|
|  | training | test | training | test |
| nMAE | 2.22 % | 2.42 % | 2.15 % | 2.14 % |
| nME | -0.05 % | -0.17 % | -0.11 % | -0.10 % |
| nRMSE | 4.02 % | 4.21 % | 3.69 % | 3.66 % |
| R2-score | 0.99 | 0.99 | 0.99 | 0.99 |

which equals the proposed wind farm power model, but without convolution layers. It has the same set of input features, but only for time step $t_0$. Comparing the performance metrics of the wind farm power model proposed in this work (Table 5) with those of the two baselines models (Table 6), one can see, for example, that for the test data the nMAE is reduced with about 47% and 18% respectively (for both wind farms). Thus, both the usage of additional input parameters (i.a. turbulence intensity, air density and setpoint of farm power controller) and information from the preceding time period (30 minutes) improve the performance of the farm power model significantly. Note that none of the the baseline models can capture the dynamics caused by variable wind inflow.

**Table 6.** Performance metrics for two baseline deep learning power models for both Wind Farm 1 and Wind Farm 2

|  | FNN($v_0, \theta_0$) | | | | FNN(all input features at $t_0$) | | | |
|---|---|---|---|---|---|---|---|---|
|  | Wind Farm 1 | | Wind Farm 2 | | Wind Farm 1 | | Wind Farm 2 | |
|  | training | test | training | test | training | test | training | test |
| nMAE | 4.16% | 4.58% | 4.48% | 4.06% | 2.38% | 2.94% | 2.54% | 2.59% |
| nME | 0.95% | 1.04% | 1.48% | 1.14% | 0.06% | -0.07% | -0.04% | 0.00% |
| nRMSE | 7.84% | 8.58% | 9.57% | 8.45% | 4.36% | 5.19% | 4.43% | 4.54% |
| R2-score | 0.95 | 0.94 | 0.92 | 0.94 | 0.99 | 0.98 | 0.98 | 0.98 |

Figures 14 and 15 give some insight in the prediction error for individual test data points. It can be seen that the prediction error is the smallest for conditions where the farm power is close to 0% or 100% of the installed capacity. Indeed, these regions comprise respectively many data points with wind speeds below the turbine cut-in wind speed and wind speeds above the rated turbine wind speed, where the power curve of the turbines is relatively flat.

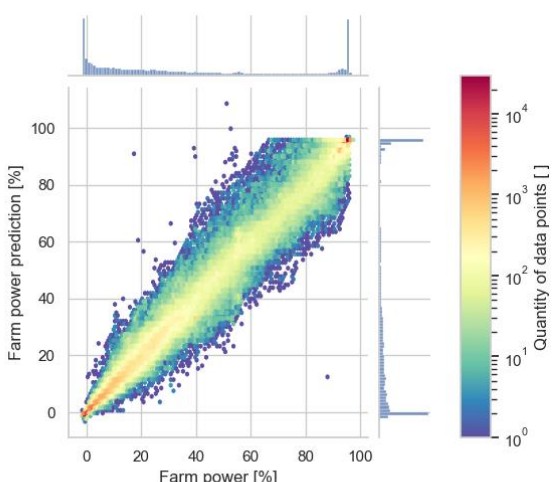
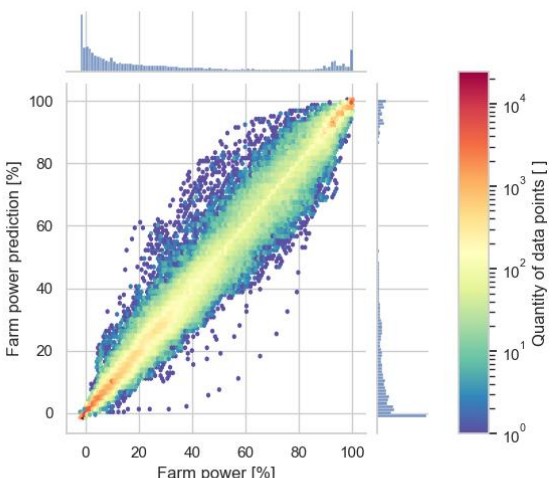

**Figure 14.** Relation between predicted and true farm power for all data points of the test data set for Wind Farm 1

**Figure 15.** Relation between predicted and true farm power for all data points of the test data set for Wind Farm 2

### 3.1.4 Validation time sequence

Figure 16 shows the predicted and true farm power for a 60-hour time sequence of validation data for Wind Farm 1, as well as the predicted confidence intervals. The confidence intervals are the smallest for high wind speeds resulting in maximum
farm power and for low wind speeds with a farm power close to 0 MW. The uncertainty appears to be the highest when the farm power is fluctuating and peaking heavily. In contrast, for long continuous power increases or decreases, the predicted uncertainty of the model appears to be relatively low. For this time sequence, 69.0% of the of the 1-minute true farm power data points lay within the 68.3% confidence interval and 97.5% of the data points lay within the 95% confidence interval (when ignoring the true farm data points at maximum and zero farm power, where the absolute prediction error is negligible).
Figure 17 shows the predicted and true farm power for a time sequence of test data during which the farm power controller of Wind Farm 2 is actively curtailing the farm power (i.e. the set point of the power controller is smaller than 100%). The uncertainty of the model appears to be very low when the wind speed is largely sufficient to attain the power set point.

### 3.1.5 Sensitivity analysis

Figures 18 to 23 illustrate the sensitivity of the farm power curve generated by the farm power model for each of the input
parameters. In each of the plots, the farm power is shown as a function of the wind speed. The other input parameters are predicted by the corresponding auxiliary models (see §2.3.4) based on the wind speed and a specific wind direction. One single

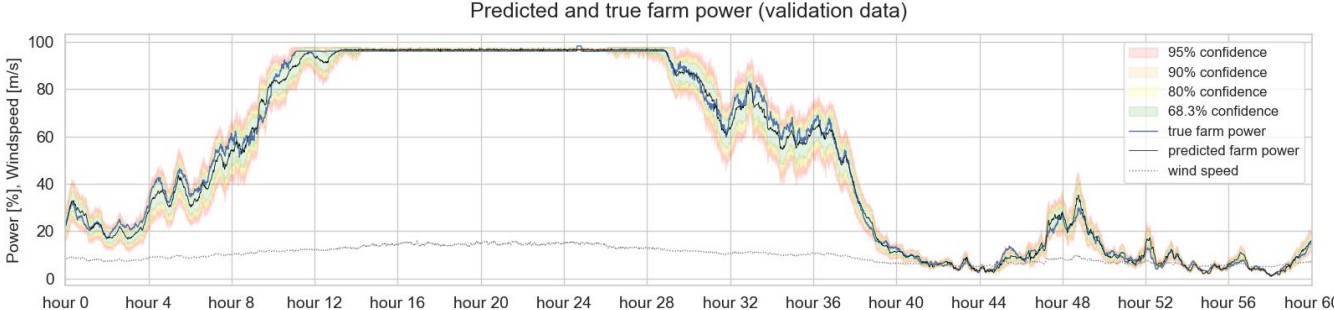

**Figure 16.** Predicted farm power and uncertainty intervals for a 60-hour time sequence of validation data (Wind Farm 1)

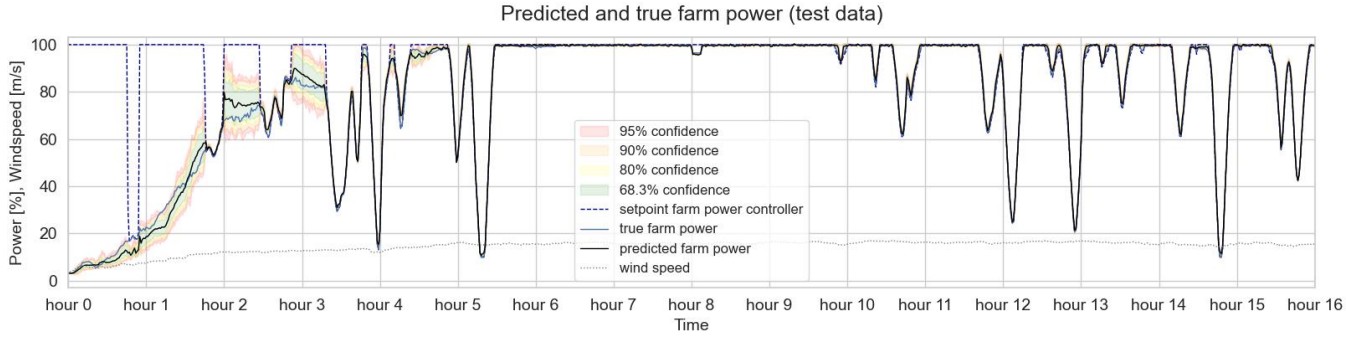

**Figure 17.** Predicted farm power and uncertainty intervals for a 16-hour time sequence of test data with active power control (Wind Farm 2)

input parameter is then adapted slightly in order to analyse the resulting impact on the predicted farm power. Each simulation is a steady-state simulation, i.e. the value of each of the input parameter sequences is constant in time.

Figures 18 to 20 show that the models predict a higher farm power for respectively an increased turbulence intensity, wind direction variance and air density. This confirms what can be seen in figures 8, 10 and 12. Figure 21 shows the predicted power for Wind Farm 1 for two different wind directions. The produced power for wind direction 240° is lower than for wind direction 180°. This difference is due to farm-internal wake (see also section 3.2). Indeed, wind direction 240° is parallel with a long cross-section of Wind Farm 1. In contrast, wind direction 180° is slightly off the short cross-section of the wind farm. For this wind direction, the internal wake is thus minimal. Figure 22 shows the predicted power reduction for Wind Farm 2 when respectively one and two turbines are stopped. As mentioned in §3.1.1, the maximum power of this wind farm is limited by its farm controller. As can be seen in Figure 22, at high wind speeds this maximum power limit can almost be reached with one turbine at standstill. Figure 23 shows the farm power for Wind Farm 2 in case the set point of the farm power controller equals 30%, for the wind directions 325° and 260°. For low wind speeds, the farm power for wind direction 260° is lower than for 325°, because of the higher farm-internal wake.

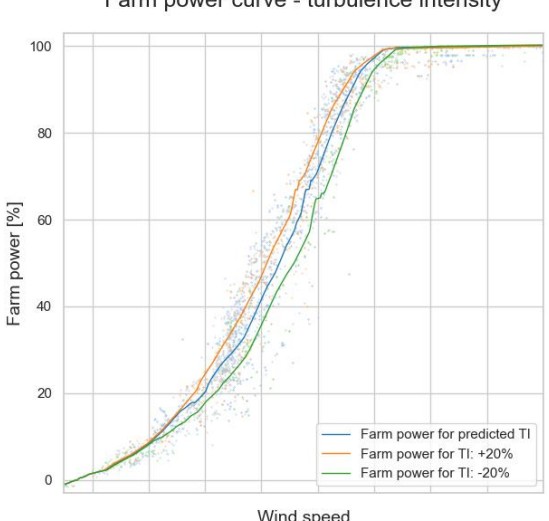

**Figure 18.** Predicted farm power as a function of inflow wind speed and wind turbulence intensity (TI) (Wind Farm 2)

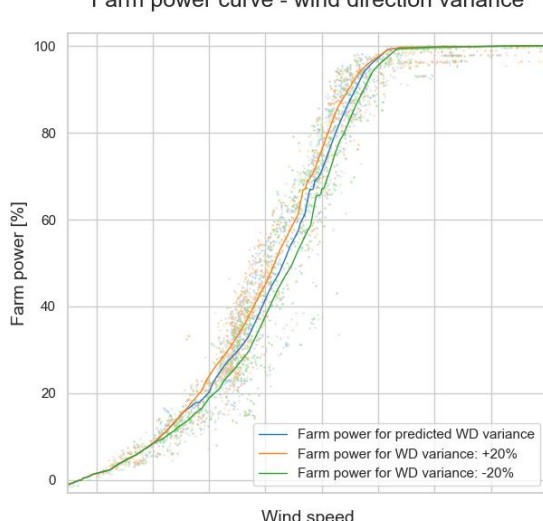

**Figure 19.** Predicted farm power as a function of inflow wind speed and wind direction (WD) variance (Wind Farm 2)

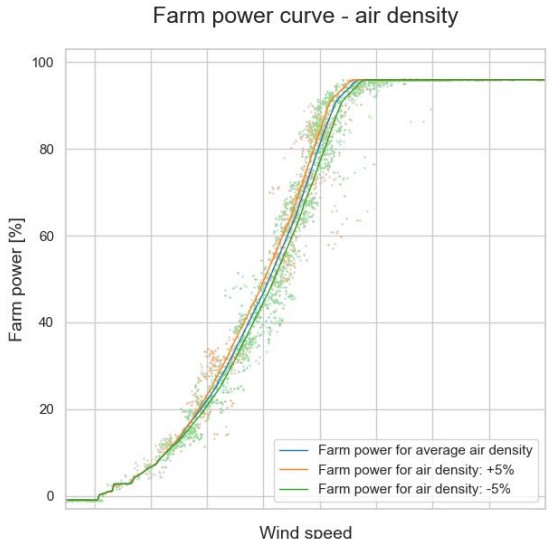

**Figure 20.** Predicted farm power as a function of inflow wind speed and air density (Wind Farm 1)

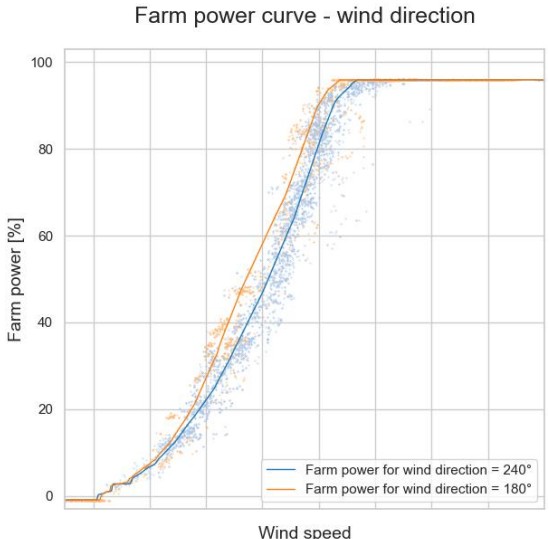

**Figure 21.** Predicted farm power in function of inflow wind speed and wind direction (Wind Farm 1)

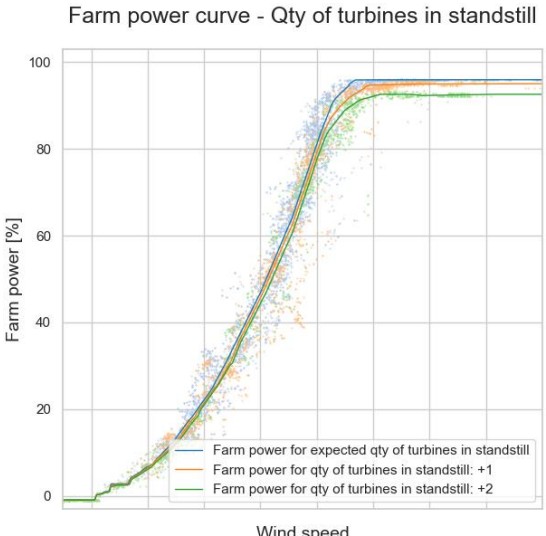

**Figure 22.** Predicted farm power in function inflow of wind speed and the quantity of turbines in standstill (Wind Farm 1)

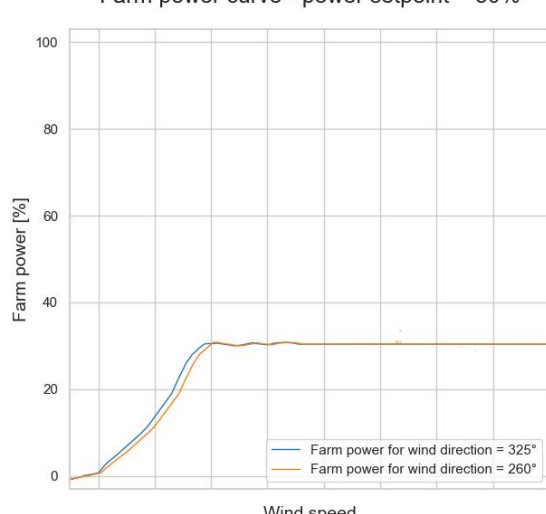

**Figure 23.** Predicted farm power in function of inflow wind speed and direction, with the farm power controller set point equal to 30% (Wind Farm 2)

### 3.1.6 Farm power dynamics

All power predictions presented in §3.1.5 are for steady-state conditions, i.e. all input parameters of the model are constant, at least during the last thirty-one 1-minute time steps $t_i \in [t_0, t_{-1}, ..., t_{-30}]$, which are input features of the farm power prediction model. As explained in §2.3.1, the structure of the neural network of the farm model has been chosen specifically to be able to capture temporal variations of the inflow wind characteristics. In this section, the farm power is predicted for wind speeds that fluctuate over time. The objective is twofold: firstly, to assess the ability of the model to predict consistent and physically meaningful results under these dynamic conditions, and secondly, to gain insights into how wind speed variations impact farm power production, what cannot be modeled by traditional steady-state power models. Two types of wind speed variations are tested: linear wind speed ramps and sinusoidal varying wind speeds with different frequencies.

Figure 24 shows power predictions for Wind Farm 1 for the same wind directions as shown in figure 21 (i.e. 150° and 240°). However, in addition to the two power curves for constant wind speeds ($v_i = v_0$ for $t_i \in [t_0, t_{-1}, ..., t_{-30}]$), power predictions are shown for the cases with a linear wind speed increase and decrease with a change rate of 0.05 m/s per minute (i.e. resp. $v_i = v_{i-1} + 0.05$ m/s and $v_i = v_{i-1} - 0.05$ m/s). It can be seen that in case of increasing wind speed (dashed lines), the predicted farm power is lower. In contrast, for decreasing wind speed (dotted lines), the predicted power is higher. Indeed, if at the inflow side of the wind farm the wind speed is increasing, this means that downstream in the wind farm the wind speed is still lower than at the inflow side, which results in a lower total farm power. As can be seen on this plot as well, this effect is larger for wind direction 240° (blue arrows) than for wind direction 180° (orange arrows). This is because the cross-section of the wind farm in direction 240° is longer than in wind direction 180°. Consequently, changed wind speeds need more time to reach

the downstream turbines. In addition, on the plot it can be seen that there is a hysteresis for the start-up and shut-down of the turbines around the cut-in wind speed. Note that for static wind farm power models, in Figure 24, all curves for wind direction 150° would be identical and all curves for wind direction 240° would be identical.

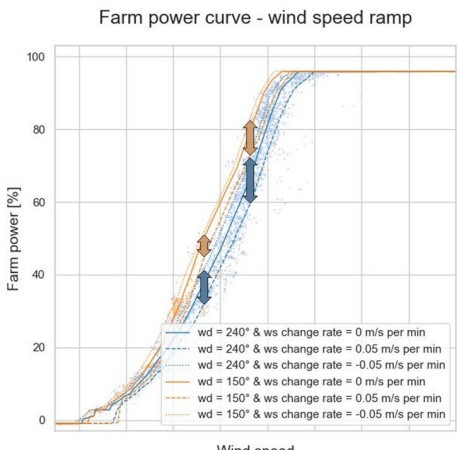

**Figure 24.** Predicted farm power in function of inflow wind speed (ws) with a linearly increasing and decreasing speed with a change rate of 0.05 m/s per minute, for wind directions (wd) 240° and 150° (Wind Farm 1).

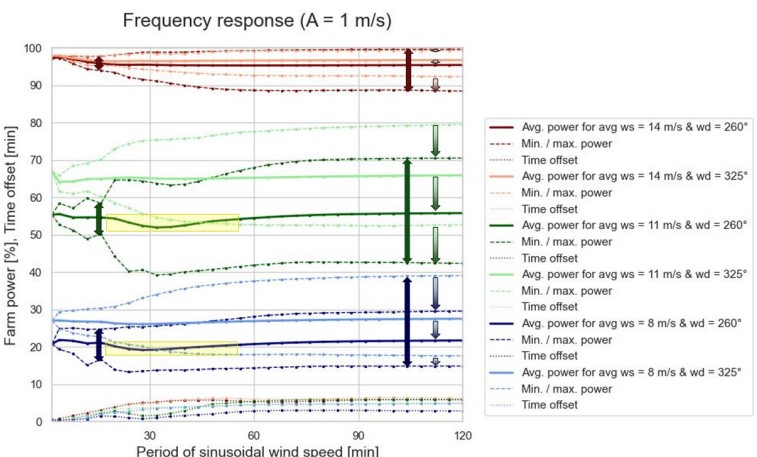

**Figure 25.** Frequency response of predicted farm power in case of a wind inflow with a sinusoidal oscillating wind speed component with an amplitude of 1 m/s. The plots show the average, maximum and minimum power of the predicted oscillating farm power in function of the period of the sinusoidal component of the wind speed. In addition, also the time offset between the oscillating farm power and wind speed is shown. The color of the curves indicate the average wind speed (resp. 14 m/s, 11 m/s and 8 m/s) and the wind direction (260° and 325°) (Wind Farm 2).

Figure 25 shows the frequency response of the power model for Wind Farm 2. The inflow wind speed is simulated as a constant average speed $V$ superposed with a sinusoidal component with an amplitude of 1 m/s ($v_i = V + sin(\frac{2\pi}{T} \times t_i)$). Simulations were run for a period $T$ of the sinusoidal component equal to 2 minutes up to 120 minutes ($T$ = 2, 4, 8, 12, 16, 20, ..., 120 minutes). This has been done for different average wind speeds ($V$ = 14 m/s, 11 m/s and 8 m/s) and wind directions (260° and 325°). The curves show the average, maximum and minimum values of the resulting oscillating farm power (solid and two dashed lines), as well as the time delay between the sinusoidal wind speed component and the oscillating farm power (dotted lines).

As an oscillation period of 120 minutes is much longer than the time needed for the wind to cross the complete wind farm, this is a quasi-static wind condition. For smaller oscillation periods, the frequency of the wind oscillations is higher. However, for $T$ = 2 minutes, taking into account the 1-minute time step, the wind speed is again constant (because $sin(\frac{2\pi}{T} \times t_i) = sin(\frac{2\pi}{2} \times n) = 0$).

As can be seen on the plot, for a specific wind speed and oscillation frequency, the average, maximum and minimum farm power for wind direction 260° (with high wake) is always lower than for wind direction 325° (with less wake) (see downward

arrows). For faster fluctuating wind speeds (i.e. with shorter oscillation period), the amplitude of the farm power fluctuation decreases (see bidirectional arrows). In addition, in case of wind conditions with a high farm-internal wake, there appears to be a decrease of the average farm power (see yellow markings). For small oscillation periods below 15 minutes, the average power is increasing again, converging back to the same value as for quasi-static wind conditions.

For short oscillation periods and consequently small wavelengths of the spatial wind speed distribution, the time delay of the farm power is converging to zero. Indeed, the time delay cannot be longer than the oscillation period of the sinusoidal wind speed component. For longer oscillation periods, the time offset converges to a constant value. Logically, the time delay will never be higher than the time needed by the wind to cross the complete farm.

For short oscillation periods below 15 minutes, the wavelength of the wind speed oscillation is getting smaller than the cross-section of the farm. This may cause the important decrease of the amplitude of the farm power oscillation. This might also explain the jigsaw shape of the maximum and minimum power in these conditions. To analyze these effects in more detail, a wind farm model could be used that does not only predict the total farm power, but also the power production of each individual turbine.

Note that for static wind farm power models, all curves in Figure 25 would be horizontal lines and be equal to the values for the longest shown oscillation period ($T = 120$ minutes).

The analysis in this section is a qualitative analysis. As the simulations are done with fictive wind speeds, there are no ground-true values for the farm power production to compare with. Note, however, that the performance metrics presented in Table 5 are for measured wind data and cover thus wind speed variations like they occur in reality. These performance metrics cover thus the dynamic behavior of the farm power model.

## 3.2   Farm-internal wake

As already shown in Figure 21, the farm power production depends on the wind direction due to the difference in wake loss. In order to calculate the farm-internal wake in absolute terms, an ML-model has been established for a single turbine, as described in §2.3.2. By subtracting the power predicted by the farm power model from the predicted turbine power under identical wind conditions (multiplied with the number of turbines in the farm), the farm-internal wake effect can be isolated from other influences on the farm power.

Figure 26 shows the measured farm power for Wind Farm 1, as a function of the wind direction and wind speed. Figure 27 shows the subset of these data points for which the set point of the farm power controller is equal to 100% and at maximum one turbine is at standstill. Furthermore, the scope of the plot has been limited to the wind speed range for which wake is most predominant. Figure 28 shows the corresponding predicted farm power by the farm model for steady-state wind inflow (during 30 minutes). It can be seen that for the directions west-southwest and east-northeast for a given wind speed, the farm power is lower than for other wind directions. This can be seen more clearly after subtraction from the predicted turbine power. Figure 29 shows the power loss due to internal wake as a percentage of the installed capacity of the farm. The maximum internal wake loss is about 30%. The reason for the high wake in the west-southwest and east-northeast directions, is that these directions are parallel to the long axis of the wind farm, with multiple turbines positioned after each other. For wind speeds above 13 m/s,

the power loss due to wake approaches 0 MW. This is because at such high wind speeds there is sufficient energy in the wind, and consequently each turbine in the farm can produce sufficient power so that the maximum farm power is reached.

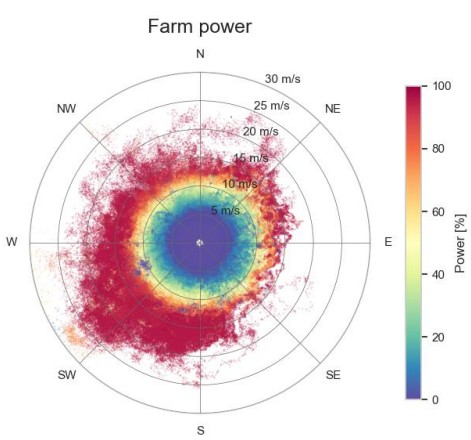

**Figure 26.** Farm power measurement data points in function of wind speed and direction (Wind Farm 1)

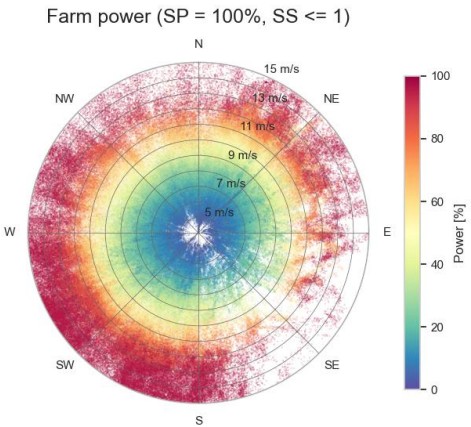

**Figure 27.** Subset of farm power measurement data points in function of wind speed and direction, with farm power control set point equal to 100% and quantity of turbines in standstill equal to 0 or 1 (Wind Farm 1)

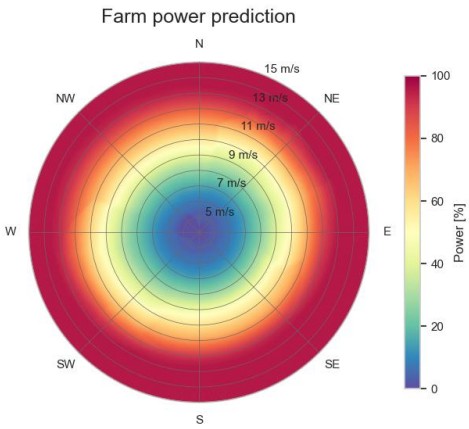

**Figure 28.** Predicted farm power in function of wind speed and direction, for constant wind conditions, farm power control set point equal to 100% and with all turbines in operation (Wind Farm 1)

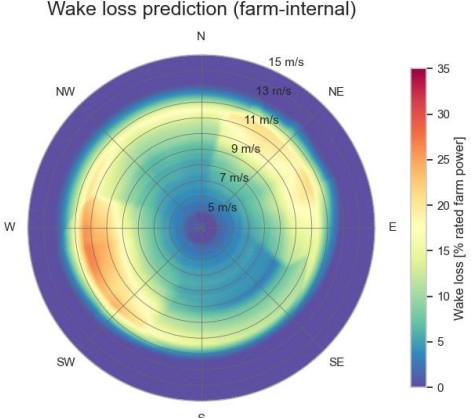

**Figure 29.** Predicted farm-internal wake in function of wind speed and direction, for constant wind conditions, farm power control set point equal to 100% and with all turbines in operation (Wind Farm 1)

### 3.3 Power forecasting based on weather forecasts

The farm power prediction models presented in sections §3.1 and §3.2, predict the farm power based on measurement data of the wind inflow and some other parameters of the farm. In order to forecast the farm power in the future with these farm models, forecasts of the wind speed and direction (and if available also the other wind characteristics) are required.

Weather forecasts of different providers may differ among each other as they can be based on different weather models and data. Furthermore, they typically do not take (accurately) local effects into account, like for example neighboring wind farms. These may cause a reduction of the wind speed, an increase of the wind turbulence intensity and a redirection of the wind due to wind blockage. Also coastal effects may have a significant impact on the wind speed and direction.

Figure 30 shows the correction factor to be applied to the wind speed forecasts of a specific weather forecast provider for Wind Farm 1, depending on the forecast wind speed and direction. This correction factor has been calculated by mapping historical wind speed forecasts (those with the shortest lead time) from that weather forecast provider to the corresponding measured inflow wind speed of that farm. As can be seen in the figure, for wind directions between north-northwest and north-northeast, the wind speeds predicted by the correction model are only about 80% of the forecast wind speeds. The reason is that in that upstream direction many wind farms are located in immediate proximity. For wind speeds below 5 m/s, the correction factor is higher than one. This is due to the fact that the wind measurements on the turbines are not well calibrated and are over-estimating these low wind speeds. For such wind speeds below the cut-in wind speed, turbines are shut off anyway.

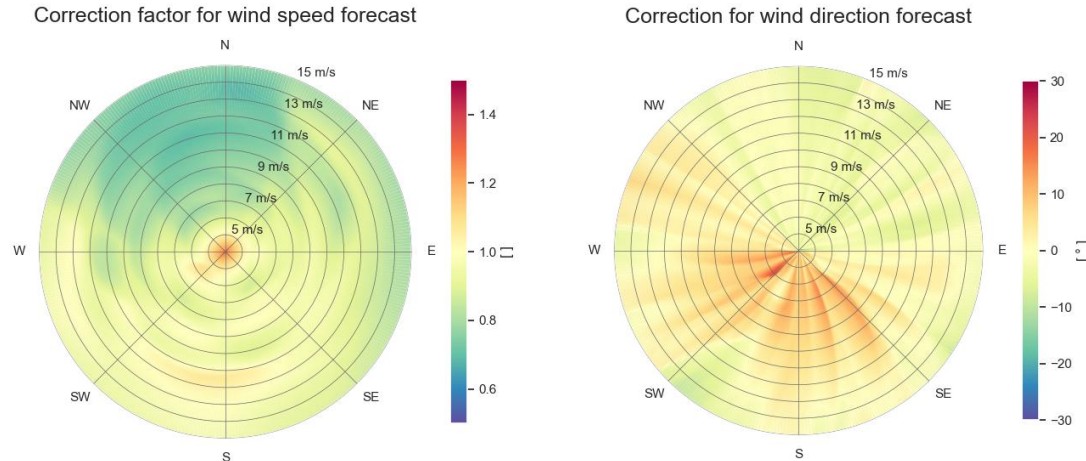

**Figure 30.** Correction factor for the wind speed forecasts of a weather forecast service for Wind Farm 1

**Figure 31.** Correction for the wind direction forecasts of a weather forecast service for Wind Farm 1

In Figure 31, it can be seen that for forecast wind directions in the sector from northwest, over south to east, the wind directions are in reality about 10 to 20 degrees higher, meaning that air flows coming from these directions are deflected by about 10 to 20 degrees in clockwise direction (compared to the values forecasted by the weather forecast provider). In contrast,

air flow coming from the sector east to northwest, are deflected in anti-clockwise direction. This corresponds to the fact that Wind Farm 1 is located in the south of the wind farm cluster and the cluster has a rectangular-like shape with the long axis in northwest-southeast direction. Due to blockage of the wind by the wind farm cluster, the air flow deviates through pressure build-up in front of the cluster slightly in the direction of the outside corners of the cluster, where it can flow next to the cluster. For lower wind speeds, thus with lower momentum, this deflection appears to be sharper than for higher wind speeds.

In addition, in Figure 30 it can be seen that on average the corrected inflow wind speeds are lower than the ones forecasted by the weather service provider, also for wind directions without upstream wind farms causing wake. This may be may be partly attributable to coastal effects (the coast is the nearest in east to south directions) and partly to the blockage effect of the wind farm cluster. For the wind directions in the sector south-southwest (directed towards the corner of the wind farm cluster) slightly increased wind speeds can be observed (especially for wind speeds between 11 m/s and 12 m/s). This might be an indication of the acceleration of the air flow at the corner of the cluster occurring jointly with the deflection due to the blockage effect. However, this may also be attributable to other reasons, such as under-estimation by the weather forecast provider of wind speeds in parallel with the coastline through the narrow Strait of Dover, as weather forecast models cannot model all local effects. For forecasts from another weather forecast provider and for historical ERA5 data, similar discrepancies in wind speed and wind directions are observed, however with different biases and/or variances.

Using the complete chain of models as shown in Figure 2, starting from the models correcting the wind speed and direction forecasts, the auxiliary models to predict the wind turbulence and air density, and finally the farm power model, the farm power can be forecasted for multiple time horizons. Figure 32 shows the three-days-ahead, day-ahead and intra-day power forecasts for an eight-days sequence for Wind Farm 1. Figure 33 shows the corresponding wind speed forecasts used as inputs for the power forecasts. As weather forecasts with a shorter lead-time are usually more accurate than with longer lead-times, the resulting power forecasts are getting more accurate for shorter lead times as well.

This can be seen in figures 34 and 35, which show multiple error metrics for wind speed and direction forecasts with three different forecast horizons: intra-day (ID), day-ahead (DA) and three-days-ahead (3DA). From the bar charts it can be seen that the mean error (ME), mean average error (MAE) and root mean square error (RMSE) are higher for longer forecast horizons, whereas the R2-score decreases. It can be seen also that all error metrics for the wind speed forecasts improve significantly after application of the wind speed forecast correction model. Notice also that the mean error of the uncorrected wind speed forecasts is always larger than 0 m/s. This is probably mainly due to the fact that the speed forecasts ignore the wind speed reduction caused by the wake of the upstream wind farms. After applying the correction model to the wind speed forecasts, the mean error is decreased to around 0 m/s.

In Figure 36, the error metrics of the corresponding wind farm power forecasts are shown. For each forecast, the same wind farm power model has been used. For the orange bars, the uncorrected wind speed and direction forecasts have been used, whereas for the green bars, first the wind speed and direction correction models have been applied. The blue bars indicate the error metrics for the power predictions of the wind farm power model based on the actual wind inflow measurements (see Table 5, test data set of Wind Farm 1). It can be seen that the application of the wind correcting models improve the farm power forecasts significantly. Note also that without the weather forecast correction models, the average wind farm power is

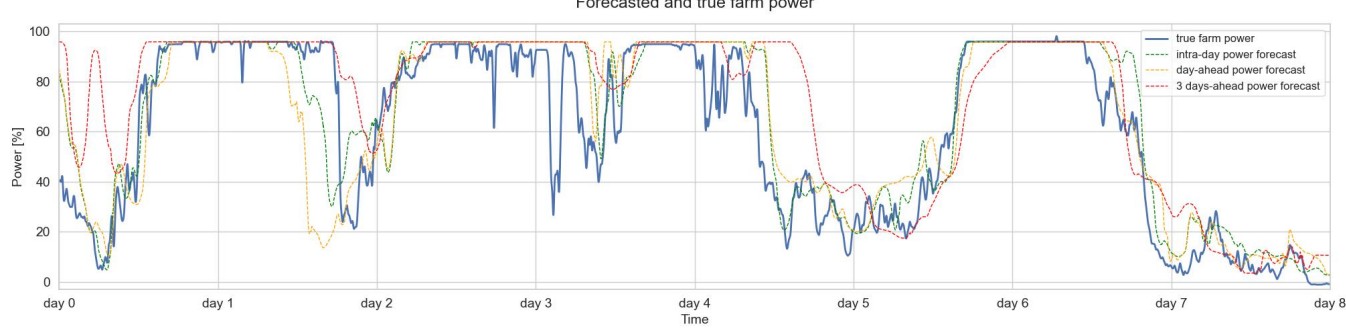

**Figure 32.** Power forecasts for Wind Farm 1 based on wind forecasts with different forecast horizons for a time sequence of 8 days

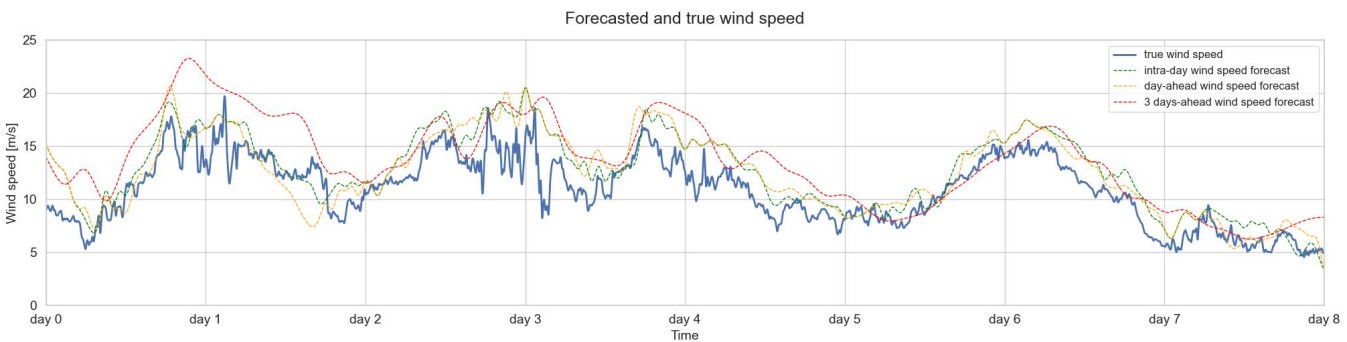

**Figure 33.** Wind speed forecasts with different forecast horizons (from a third-party weather forecasting provider)

overestimated largely. This is not the case anymore after application of the weather forecast correction models. In addition, in the case of day-ahead forecasting, the MAE and RMSE are both reduced with about 30%. For three-days-ahead forecasting, the relative improvements are smaller, but still significant.

Remark that a wind speed error of only 1 m/s represents about 12.5% of the range between the cut-in and rated wind speed of a turbine ($\sim$4 m/s to $\sim$12 m/s). Consequently, a wind speed forecasting error of that magnitude can result into a large farm power forecasting error, larger than the inaccuracy inherent to the farm power model itself. The relatively large errors in the weather forecasts should be no surprise, as the weather forecasts used in this example have a time resolution of only a 1-hour (to be compared to the 1-minute resolution of the used SCADA data and wind farm power model). Moreover, day-ahead forecasts have a forecasting lead time of up to 37 hours (from day-ahead 11 a.m. until 12 p.m.). In addition, the processing time for making these weather forecasts may take up to 6 hours, so that the day-ahead forecasts may be based on data from 43 hours ahead. The intra-day forecasts used in this example, can have a lead time of up to half a day, because the weather forecast data available for this work is updated only twice per day. Using more accurate weather forecasts updated more frequently and with a shorter time resolution will result in better farm power forecasts. One could argue that for the specific application of long-term wind farm power forecasting, using an accurate wind farm power prediction model with such short time resolution

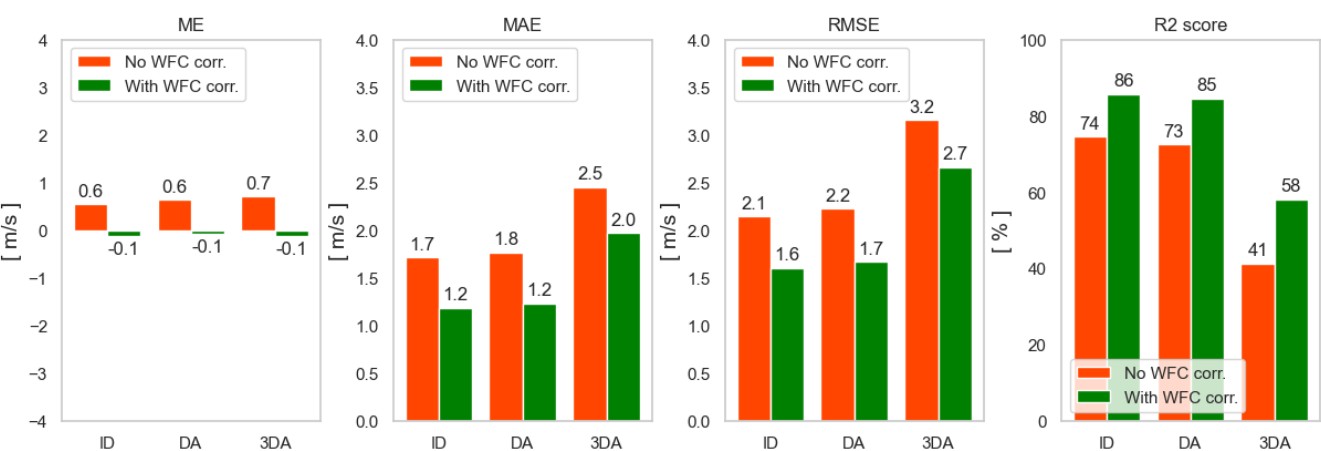

**Figure 34.** Error metrics for intra-day (ID), day-ahead (DA) and three-days-ahead (3DA) wind speed forecasts for Wind Farm 1. The orange bars are the metrics for the wind speed forecasts of a commercial weather forecast provider. The green bars are the metrics after application of the weather forecast correction (WFC) model.

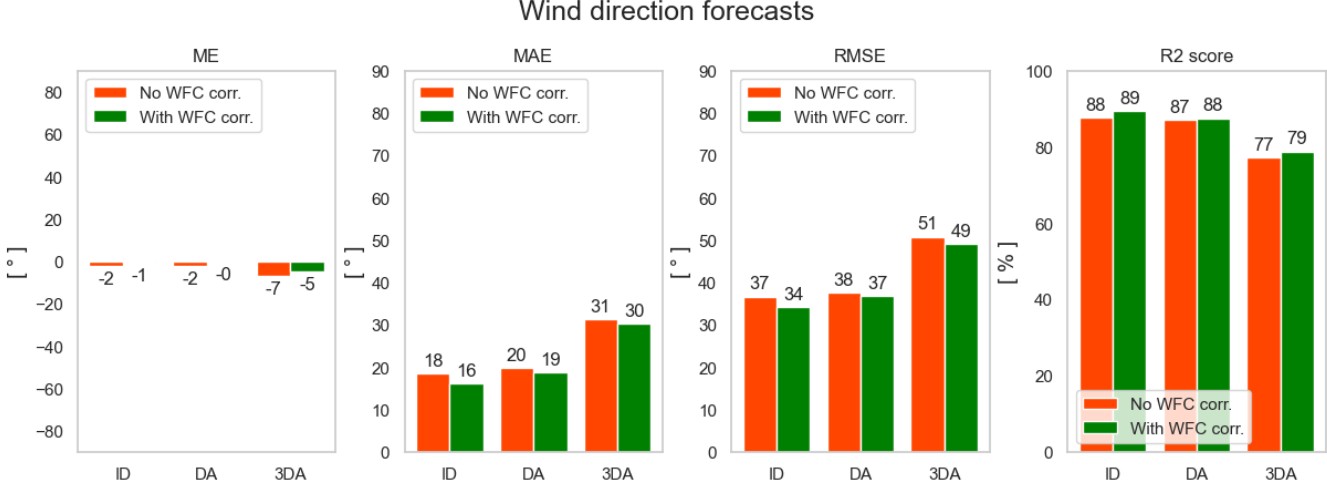

**Figure 35.** Error metrics for intra-day (ID), day-ahead (DA) and three-days-ahead (3DA) wind direction forecasts for Wind Farm 1. The orange bars are the metrics for the wind direction forecasts of a commercial weather forecast provider. The green bars are the metrics after application of the weather forecast correction (WFC) model.

(1 minute) adds little value. It should be reminded, however, that if for example a 1-hour time resolution would be used for the training of the wind farm power model, the behavior of the wind farm power controller and the temporal dynamics of inflow

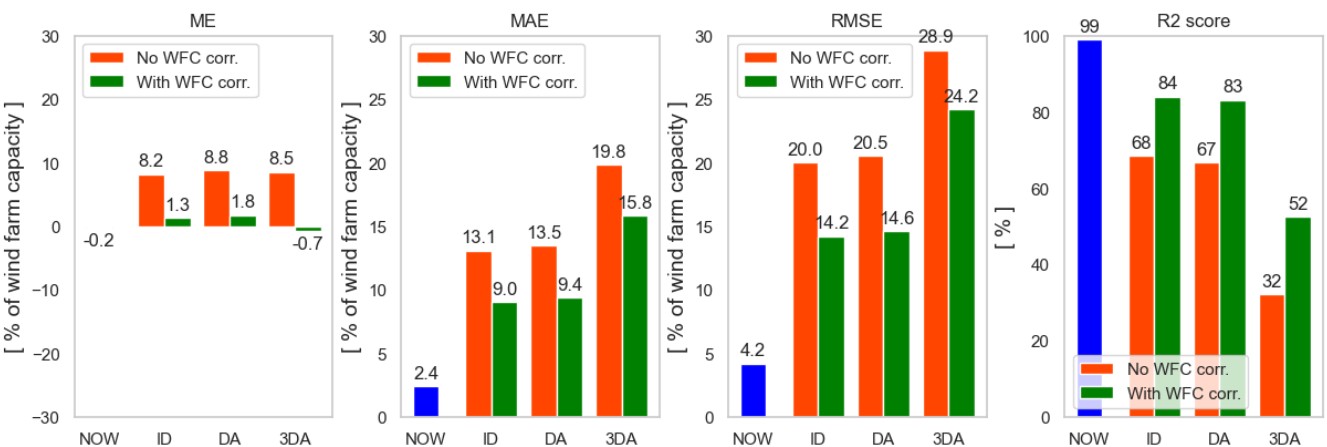

**Figure 36.** Error metrics for intra-day (ID), day-ahead (DA) and three-days-ahead (3DA) wind farm power forecasts for Wind Farm 1. The orange bars are the metrics for the wind farm power forecasts based directly on the wind speed and direction forecasts of a commercial weather forecast provider. The green bars are the metrics for the wind farm power forecasts based on the corrected wind speed and direction forecasts. The blue bars show the corresponding error metrics for the wind farm model predictions based on actual (NOW) wind inflow measurements.

wind transients would not be captured by the farm power model, which are, however, crucial for other targeted applications of the model.

### 3.4 Computing hardware & software

All computing performed for this work, including the training of the ML-models, has been performed with a standard notebook (processor: 11th Gen Intel(R) Core(TM) i7-1165G7 @ 2.80GHz, internal RAM: 16.0 GB, 64-bit operating system). All code
is written in Python, with Tensorflow as machine learning platform. Both are free open-source software. This shows that the proposed model, in contrast to high-compute models, could be run on hardware that is readily accessible to wind farm operators.

### 4 Conclusion

In the present work, a novel methodology is proposed to forecast the power production of a wind farm. The methodology is
based on a multi-component pipeline with as two main components a deep learning wind farm power model and a distinct machine learning model for integrating weather forecasts.

The proposed wind farm power model relies solely on SCADA data from the wind farm itself, which are usually available to any wind farm operator. It captures the influence of several weather parameters, including wind speed, wind direction,

turbulence intensity, wind direction variance and air density. Additionally, it captures the temporal dynamics of the wind inflow as well as the behavior of the farm power controller. Also the number of turbines that are in standstill is taken into account. Notable, the model does not only predict the farm power with a high accuracy, but generates also confidence intervals for these power predictions. Furthermore, the farm power model is capable of predicting the farm power in only a few milliseconds on PC, making it significantly faster than even low-fidelity physics-based models.

A separate deep learning model is employed to post-process the wind speed and direction forecasts from weather forecast providers. In doing so, it takes farm-external factors into account, such as wake generation by neighboring wind farms, wind farm blockage, coastal effects and possible systematic forecasting errors from the respective weather forecast providers.

The two models, i.e. the wind farm power model and the model for post-processing weather forecasts, are independent from each other, which is a major advantage, as these can be trained and maintained separately.

Furthermore, in conjunction with a data-driven turbine power model, also the farm wake losses can be predicted.

The proposed methodology has been applied to two large real-world offshore wind farms. Performance metrics affirm a significantly improved prediction accuracy compared to some baseline machine learning models. Validation sequences demonstrate in addition the reliability of the predicted confidence intervals. Sensitivity analyses, performed on each of the model's input features, yield interpretable and physically meaningful results. In addition, the prediction capability of the farm power model is demonstrated for fluctuating inflow wind speeds.

It is shown also that the application of the post-processing model to the weather forecasts, improves significantly the accuracy of the look-ahead wind farm power forecasts. Nevertheless, the accuracy for long forecast horizons remains limited predominantly by the limited accuracy of the third-party weather forecasts and not by the uncertainty inherent to the farm power model itself.

In further research, we will integrate the wind farm power prediction models as digital twins into applications where their high prediction speed and the simulation of the farm power controller are crucial.

*Author contributions.* SA: conceptualization, data curation, formal analysis, investigation, methodology, software, validation, visualization and writing (original draft preparation, review and editing); TV and PJD: supervision and paper reviewing; AN: Funding acquisition; JH: Funding acquisition, supervision and paper reviewing.

*Competing interests.* The authors declare that they have no competing interests.

*Acknowledgements.* This research was supported by the Flemish Government (AI Research Program) and financially supported by the Energy Transition Fund projects Poseidon and Beforecast.

## Appendix A: Hyper-parameters of ML models

In Table A1, the hyper-parameters of the wind farm power model (§2.3.1) are listed. Note that the hyper-parameters for the two wind farms are identical, except for the quantity of input features, as wind Farm 2 has as additional input feature: "active power capability of the farm" (§2.1.2).

**Table A1.** Hyper-parameters of wind farm power model

| Parameter | Value |
| --- | :---: |
| Quantity of input feature sequences ($K$) | 8 (Wind farm 1), 9 (Wind farm 2) |
| Quantity of additional input features ($L$) | 2 (Wind farm 1), 3 (Wind farm 2) |
| Quantity of convolution filters ($c_1, c_2$) | 8 |
| Kernel size of convolution filters | 3 |
| Stride of convolution filters | 1 |
| Quantity of units in dense layers ($n_1, n_2, n_3$) | 128, 256, 256 |
| Activation function in dense layers | ReLU |
| Activation function in output layer | linear |
| Dropout rate | 0.10 |
| Optimizer | Adam |
| Loss function | MAE |
| Initial learning rate | 0.0001 |
| Learning rate reducing factor | 0.99 |
| Minimum learning rate | 0.00001 |

In Table A2, the hyper-parameters of the wind speed and direction forecast mapping models (§2.3.3) are listed. Note that for the two models, the same hyper-parameters are used.

**Table A2.** Hyper-parameters of weather forecast mapping models

| Parameter | Value |
| --- | --- |
| Quantity of units in dense layers ($n_1, n_2, n_3$) | 16, 32, 32 |
| Activation function in dense layers | ReLU |
| Activation function in output layer | linear |
| Dropout rate | 0.25 |
| Optimizer | Adam |
| Loss function | MAE |
| Initial learning rate | 0.001 |
| Learning rate reducing factor | 0.99 |
| Minimum learning rate | 0.00001 |

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
