# Peer review of "Modular deep learning approach for wind farm power forecasting and wake loss prediction"

_Wind Energy Science, 2024_

## Author Comment (AC1)

**Reply to the reviewers**

We would like to thank all reviewers for their thorough analysis of our manuscript and their constructive comments. We have carefully considered each comment and provide below our answers. We will revise our manuscript accordingly and believe that the revision will address all reviewers' concerns. We are looking forward to hearing your feedback.

**Referee comments 1**

https://doi.org/10.5194/wes-2024-94-RC1

*Referee comment: In this work, the authors lay out a multi-part modeling framework for making estimates of real-time power output and future projections of the performance of a real farm. They construct a series of deep learning models to map from forecasted wind conditions to real farm-observed conditions, through to a fast, accurate, and uncertainty-imbued power estimate for the farm. The result is an exemplary application of machine learning techniques to a relevant wind engineering problem that is immediately useful to farm operators.*

**Response of authors**: Thank you for this insightful summary of our paper.

*Referee comment: I have two major criticisms of the manuscript, primarily centered on exposition of the methods of the work. In the wind farm control model, some clarity could be added in consideration of the intended audience of WES, who are not AI/ML users. A more circumspect description of terms including but not limited to, "convolution branches", "feed-forward neural network", "dropout layers", and "dense layers", would be useful to allow the audience-- myself included-- view the machine learning aspects of the work on their merits rather than simply as a black box.*

**Response of authors**: Thank you for the suggestion. We will introduce a basic description of the used AI terminology in the updated manuscript.

*Referee comment: Additionally, the exposition of the weather forecasting methods is sparse and should be significantly expanded. Because it holds a key to the application of the work in this manuscript, it should be clear how it works and what is strengths and weaknesses are, especially in light of the results for lookahead forecasting.*

**Response of authors**: We agree that a more expanded discussion on the weather forecast mapping would improve the manuscript, as it indeed has a significant impact on the accuracy of lookahead forecasting. We will add a description of the implementation of the wind forecast mapping models. Moreover, we will add metrics that quantify the improvement in accuracy.

*Referee comment: I would also like to see more analysis of the results. The model clearly performs well to make estimates of the farm power in static wind conditions. The results in dynamic conditions are less clear to interpret, and additional text and clarification of the plots in Figures 21 and 22 would be appreciated. This seems to be a major benefit that can be conferred by this approach and thus more clarity around this area would be very valuable to the end result. Moreover, this would couple with*

*better exposition of the forecasting efforts such that together they can be used to understand where and how this approach leads to errors in forecasted farm power.*

**Response of authors**: Thank you for the suggestion.

You mention that the model performs well to make estimates of the farm power in *static* wind conditions. However, we want to emphasize that the performance metrics presented in §3.1.2 (Table 4) are for real weather conditions. These metrics cover thus non-static wind conditions, like they occur in in reality. The improvement of these performance metrics compared to those of the equivalent wind farm model that does not take into account wind inflow variations, is significant: 18% reduction of the mean average error. This can be seen by comparing the error metrics shown in Table 5 in column "FNN(all input features at $t_0$)" with the error metrics shown in Table 4. Indeed, when a model gets only input data from one single time step $t_0$, it cannot take into account temporal inflow wind variations, as it has no information about it. Such model thus predict the same farm power in static and dynamic wind conditions.

Figures 21 and 22 show results that can only be interpreted qualitatively, as the plots are results of simulations based on virtual data which does not occur so in reality and for which there are thus no ground-truth farm power values to compare with. These results show, however, a behavior that fits to what one may expect and is physically meaningful. As mentioned in the manuscript, we believe that further research could be done on this topic, but we believe that that should be part of a separate work.

We will add additional clarifications to figures 21 and 22. Note that, if the model would *not* capture dynamic wind behavior, figures 21 and 22 would be quite trivial. Indeed, in Figure 21 all blue lines would be identical and all orange lines would be identical. In Figure 25, all curves would be independent of the frequency of the wind speed oscillations and, consequently, be horizontal and equal to the values of the slowest shown oscillation period (i.e. 120 minutes).

*I have a few additional small technical notes, which are addressed below by line number:*

- **Referee comment**: *177: the heuristic sorting algorithm for turbine selection is slightly unclear and also possibly could be effected by flow heterogeneity, blockage, or terrain; please clarify the choice of this algorithm versus purely geographic sorting, etc.*

  **Response of authors**: The used algorithm is in fact a geographic sorting algorithm on which some corrections are done. These corrections were added to be able to consider the aggregated wind inflow measurements as "free flow" wind, which is used typically in simulation tools for wind farm power and wakes, such as FLORIS. In addition, this makes it possible to separate farm-external wake from farm-internal wake, as discussed in the manuscript. We will clarify this in the manuscript and refer to our method as a corrected geographic sorting algorithm.

- **Referee comment**: *216: the weather forecast data, like the weather forecast modeling, lacks clarity of what it contains and how it is used*

  **Response of authors**: Thank you for the suggestion. As mentioned in the manuscript, wind speed and wind direction forecasts with an 1-hour time resolution are used as an input. We will provide more information on this data set, e.g., the location and heights used for the weather forecasts. We will elaborate further how this data set is used in the section about the weather forecast mapping models.

- **Referee comment**: *239: be careful with the use of "instantaneous" response variables, they can often have significant inertial effects that should be justified*

**Response of authors**: Indeed, the word "instantaneous " may be a too stringent formulation. The wind farm power model has a time step of 1 minute. With "instantaneous" is meant a response within that order of magnitude (in contrast to other input variables which have a response time which is significantly longer). We will clarify this in the revised manuscript and may opt for a more appropriate wording.

- *Referee comment*: ~370: the frequency of wind resource conditions is unclear, and this passes through to understanding how frequent high TI/high wind direction variance conditions occur in e.g. Figure 6; consider highlighting wind condition frequency in some way

  **Response of authors**: Indeed, the frequency of the wind resource conditions is not shown. We will add a wind rose for each wind farm, as well as histograms for TI, wind direction variance and air density.

- *Referee comment*: 416: I suggest quantifying the quality of the CI estimates: the value should be in the 68% CI explicitly 68% of the time (rather than "most of the time") and leave the 95% CI 5% of the time (i.e. "rarely") for a well posed confidence interval

  **Response of authors**: We will quantify the quality of the CI estimate.

*Referee comment: Overall the paper is of high quality and in my opinion is an exemplary application of ML to a relevant problem to wind energy researchers and operators.*

**Response of authors**: Thank you for your appreciation of our work.

**Referee comments 2**

https://doi.org/10.5194/wes-2024-94-RC2

*Referee comment: This paper is highly detailed and well-structured, presenting comprehensive approach to wind farm power forecasting using modular deep learning models. The authors effectively utilize a robust data foundation, drawing from extensive SCADA data, to build modular machine learning models capable of predicting wind farm power also accounting for wake effects. The integration of individual turbine models into a wind farm model is a significant strength of this work, enabling a precise representation of wind farm dynamics. Additionally, the thorough analysis of various factors that influence wind farm power, such as wake effects, turbulence, and site-specific conditions, adds substantial value to the research and offers an insightful perspective on the complex influences impacting power forecasting.*

**Response of authors**: Thank you for your appreciation of our paper.

*Referee comment: While the methodology is technically sound, the high level of complexity could pose challenges for practical deployment and maintenance in real-world wind farm operations. A discussion on the practical implications of model implementation and maintenance, including possible strategies for overcoming these challenges, would be a valuable addition to enhance the model's applicability for wind farm operators.*

**Response of authors**: We agree that the practical deployment and maintenance of ML models can be challenging. However, we believe that these challenges are not specific to the ML methodology proposed in our paper. We are afraid that adding this layer of generic information would overload the paper and could provoke more questions than answers to the majority of the readers of this journal, who are typically not AI or data experts.

We want to underline that our modular approach has as advantage that the wind farm power prediction model is independent from the weather forecast mapping models. As a consequence, changing the weather forecast service has no influence on the wind farm power model. Similarly, a change in the wind farm has no influence on the weather forecast mapping models. These models can be trained and maintained separately.

As indicated in the paper, depending on the specific objective of the wind farm operator, simplifications in the methodology are possible. For example, as shown in Figure 3, the "Wind turbine power model" is only required for quantifying the wind farm internal wake, and is thus not required to forecast the farm power. In addition, if there is no need to distinguish between external and internal wake losses, the heuristic sorting algorithm for the turbine selection for the aggregation of the wind inflow measurements (as shown in Figure 1), can be replaced by a simpler geographical sorting algorithm which is fixed and can be saved in a look-up table.

*Referee comment: The paper would benefit from including a comparison with simpler forecasting models in terms of prediction accuracy. Demonstrating the added value of this complex approach over basic models would strengthen the justification for its use, especially for readers interested in balancing accuracy with model complexity in practical applications.*

**Response of authors**: A comparison with the accuracy of simpler wind farm power models is given in Table 5 and lines 399-408. Please refer also to lines 550-561, where we emphasize that for long-term forecasting the use of a model with a 1-minute timestep adds little value. However, for other use cases, e.g. related to wind farm power controlling, a short time step is required. Remind also that, for wind

farms that apply active power control (e.g. for balancing the power generation and consumption portfolio of the farm owner, or for providing grid balancing services), the control actions have a predominant influence on the wind farm power and, consequently, on the SCADA data that is used for the training of the model. Averaging this SCADA data over longer time periods (e.g. 15 minutes) would degrade the model.

In the revised version of our manuscript, we will expand the description of the simpler baseline models and add quantitative comparisons with power forecasting that ignores the correction of weather forecasts.

*Referee comment*: *Line 91: The statement "The model proposed in this paper predicts the power of a complete wind farm as a whole" may lead to some ambiguity. Since the total power output of a wind farm is often predicted based on aggregate measurements, it would be more accurate to emphasize that this paper's contribution lies in the combination of individual turbine models with wake effects to produce an overall wind farm forecast.*

**Response of authors**: In this paper, we use indeed an individual turbine power model. However, for the prediction of the wind farm power specifically, this individual turbine model is not used. We work with aggregate measurements (refer to the heuristic turbine selection method shown in figure 1). The individual turbine model is used only to predict the power loss due to farm internal wake (see the overview flow diagram of our method in Figure 3). We apologize for the confusion and will clarify the statement mentioned above in the updated manuscript.

**Referee comments 3**

https://doi.org/10.5194/wes-2024-94-RC3

***Referee comment***:

*# General comments*

*This work proposes a new machine learning model for wind farm power forecasting. The model is able to predict the farm power and the wake losses at different time scales, by combining weather forecast services and deep neural networks with richer inputs. It is an interesting work proposing a new method to leverage both machine learning techniques and expert tools such as weather forecast services. Using Monte-Carlo dropout is a really nice touch, as it demonstrates a way of quantifying the uncertainties with ML approaches.*

**Response of authors**: Thank you for your overall appreciation of our paper.

***Referee comment***: *I have 3 major criticisms: 1) the readability is not great, 2) too much information about the ML models are missing, and 3) the experiments do not really demonstrate the added value of the proposed methodology.*
*1) The paper could generally benefit from more graphics giving an overview of the whole model, showing the different elements and their dependencies. I think you developed quite an interesting architecture, and it would be really helpful to have a single picture summering the whole process. Then, it would become much easier for the reader to know to what submodule a paragraph is refereeing to. The Figure 3 tries to do so, but I find it not clear enough and too general. At the beginning of each Section, you can introduce the different submodules you're going to explain (it is missing in 2.2 for example), etc.*

**Response of authors**: Thank you for the suggestion. Figure 3 gives indeed a general overview of the complete modular structure. We will add a more comprehensive introduction in section 2.2 and refer earlier to the overview figure. On the figure, we will add for each model the number of the chapter describing that model. We will also change the order of some subsections. We believe that, in doing so, it will be easier for the reader to know what submodule each paragraph is referring to and that the readability will be improved.

***Referee comment***: *There are some sub-Sections not at the right place: for example, train/test/validation data (2.1.5) is not part of the data sources, but is part of the experimental setting,*

**Response of authors**: Thank you for the suggestion. In our revised manuscript, we will shift this part to the section "Results", where we describe in detail how the data split is done.

*the farm internal wake loss (2.2.2), is not part of the ML models, but it is a simple formula using the outputs of some ML models if I corrected understood, etc.*

**Response of authors**: You have indeed correctly understood that the farm internal wake loss is calculated by a formula using the output of two ML models. A scalar formula can also be denoted as a model, but it is indeed more ambiguous whether a function of ML models can be seen as a ML model. We believe that the readability will be increased by first introducing all ML models and shift subsection §2.2.2 about the internal wake loss towards the end of section §2.2.

*Referee comment*: 2) You have multiple "ML models" in your proposed approach. Sometimes you describe them, sometimes you don't. For example, the ML models for mapping weather forecasts (2.2.5) is interesting: adapting a weather forecast to a specific wind farm is a great idea, but you only give 2 lines of explanation on it. And you just describe it as an "ML-model": you need to give specific algorithms and models (is it a deep neural network, is it a linear regression, etc.).

**Response of authors**: Thank you for the suggestion. We agree that the explanation on the weather forecast mapping models is indeed too short. In the revised manuscript, we will describe how we have implemented these models. Moreover, we will provide additional quantitative results of these mapping models, to underline the improved forecasting accuracy when using the weather forecast correction subblock.

*Referee comment*: More generally, try to avoid generic expressions such as "ML models", but directly use the correct and specific type of algorithm.

**Response of authors**: Thank you for the suggestion. We will address the specific models whenever applicable.

*Referee comment*: In 2.2.1, how do you go from you K branches to a feed-forward layer? You need to add more details to your explanations and figures. In the results Section, you need to give all the hyperparameters of your training (learning rate, etc.). You need to carefully describe each module of your approach and provide sufficient details to ensure your work can be reproduced.

**Response of authors**: The outputs of the convolution layers are densely connected to the following dense layer.

In our revised manuscript, we will summarize the hyperparameters (which are currently mentioned in the textual description) of the main models in tabular form and we will add some additional parameters. We will add also more details to the figures where relevant.

We believe that, our work can be reproduced, in principal. The implementation and the tuning of the different submodels may depend on the data sources that are available and on the main objective(s) of the final user.

If readers may wish to get more information about the detailed implementation, they may contact the authors.

*Referee comment*: 3) In the results, I miss 2 important messages: how you justify (quantify) the novelties of your architecture, and how do you compare yourself with other works? For example: https://iopscience.iop.org/article/10.1088/1742-6596/2767/9/092014.

**Response of authors**: The paper you refer to has been issued after the writing of our manuscript. It describes another use-case: it is predicting the power of a single turbine based on data from the other turbines of the farm. This model is not doing look-ahead forecasting and is only valid for static operating conditions. Therefore, the architecture of the models described in that paper are different. In the revised version of our manuscript, we will add it among the other papers that we have cited as papers that predict the power production of a single turbine (in contrast to the power production of a complete wind farm).

To justify and quantify the novelties of our architecture, we have compared performance metrics of our proposed wind farm power model with two other baselines that omit key functionalities of our

model: 1) the use of several of additional input features and 2) the use of timeseries as input features. Our proposed model outperforms these baselines significantly.

In the revised version of the manuscript, we will also compare the performance metrics for the lookahead power forecasting with and without weather forecast correction model. The performance improvement is significant.

We want to underline that with our work we want to focus on other advantages than only the model accuracy. We want to demonstrate a methodology which allows for maximum interpretability by wind energy professionals (see e.g. the sensitivity analysis and dynamics analysis), which can be used for different forecasting horizons, captures temporal dynamics of the wind inflow and the behavior the farm power controller. The methodology also allows to predict the wake power loss and makes it possible to train and maintain independent models individually. To our knowledge, such methodology and comprehensive analysis for wind farms has not been published yet.

*Referee comment: In 3.1.2 you introduce 2 deep neural networks, but you should introduce them in the methodology section and justify why they are good baselines. More generally, I feel that the paper lacks of pedagogical approach. You propose a complex and interesting ML based model, but you need to justify each of your implementation choices. For example: you decide to start from a simple feed-forward network, and you show its limitations. Then you propose some improvements. You use a weather forecast model, and you show that it does not perform well on a specific wind farm, then you propose an ML approach to adapt it, etc. And this way, you build in a clear way your final model. In the results, it would be interesting to quantify the impact of each submodule: the impact of separating or not the inputs depending on their direct or indirect impact on the farm power, adapting weather forecast models or not, etc. You need to compare your approach with other baselines: other ML based methods, simpler versions of your solution, adapted and non-adapted weather forecasts, etc.*

**Response of authors**: We understand the writing approach you suggest. During our work, we have effectively experimented starting from simple models to more complex models in order to improve the accuracy and interpretability of the results.

We believe that documenting all intermediate steps and all possible variations will overload this manuscript and have a negative effect on the readability. We have intentionally structured our paper to be as much as possible understandable and recognizable for wind energy professionals.

In the revised version of our manuscript, like you suggest, we will add a separate section about the baseline models. We have chosen these baseline models, in order to be able to compare the accuracy when specific key implementation choices in our proposed model are omitted. In the revised version of our manuscript, we will also add a quantitative analysis of the impact of the weather forecast mapping models.

*Referee comment: I have 2 bonus questions regarding the method used to compute upstream wind turbines. 1) As I understand, every turbine can be defined as an "upstream" turbine if it has some other turbines in its wake. Therefore, even if a turbine is in the middle of the farm, it can impact the general wind flow for more downstream machines. And I feel that considering only the first upstream machines could result in a loss of pertinent data, for some wind farm layout at least. Then, how do you justify considering only the first upstream turbines?*

**Response of authors**: You are correct that wind turbines in the middle of the farm can generated wake for turbines that are located even more downstream. This is taken into account by the wind farm power model, as it is trained with the aggregated power production of *all* turbines in the farm. The heuristic algorithm for selecting the upstream turbines is only used for determining the aggregate wind inflow

conditions into the wind farm. This wind is denoted as the free flow wind. The free flow, which is not influenced by the farm, is typically used as input for wind power simulation tools and for determining the wind resource available to the wind farm. In addition, the upstream turbines measure (in general) as first changes in the wind conditions. Moreover, the wind speed measured by a turbine in the middle of the wind farm may depend strongly on control actions of the wind farm controller, and would thus not be an independent variable. Consequently, it cannot be used to interface with the weather forecast mapping model (which should be independent from the status of the wind farm). We will clarify this in the updated revision of our manuscript.

*Referee comment: 2) Why do you always consider a fix amount of upstream turbines? I think it is related to the constraints of the deep neural network's inputs, but maybe you could specify it.*

**Response of authors**: We do not fix the amount of upstream turbines. However, in the heuristic sorting algorithm we impose a minimum of upstream turbines (denoted as $N$) to be selected. The reason for that we have described in line 186+. If, for example, only 1 single upstream turbine would be used, the aggregated farm inflow measurement would only be based on one local measurement, which may not be as representative for the wind inflow, which may be heterogeneous over a large (offshore) wind farm, as it may have a cross-section of more than 10 km.

*# Specific comments*

*Referee comment: - Please always add a comma after the "e.g." and "i.e." expressions.*

**Response of authors**: We are no sure what you mean with this comment. We will check all phrases with "e.g." and "i.e.".

*Referee comment: - Please define acronyms the first time you use it. For example, you never define "PC".*

**Response of authors**: We'll add the meaning of "PC": personal computer.

*Referee comment: - Please improve the names of your variables: it is uncommon to have 2 letters for a same variable.*

**Response of authors**: We will revise the names of the variables to have only 1 character (where possible).

*Referee comment: - Please use different notations when refereeing to time series and single time step values.*

**Response of authors**: We will revise the notation where appropriate.

*Referee comment: - Please use sub-Sections in the Introduction, having a single paragraph does not really help your readers.*

**Response of authors**: We understand that the introduction is long and comprises a lot of information. We had tried to underline the structure by formatting the keywords in italics. In the revised manuscript, we will introduce some subtitles.

***Referee comment***: - *At multiple times you use the "faster than real-time power forecasting" expression. I don't really get what you mean by that. What can be faster than a "real-time" forecast?*

**Response of authors**: We mean that the predictions can be performed faster than that the events are happening in the real world. Typically, simulation runs of physics-based simulation tools are slower than the wind events are evolving in the real world. For example, large-eddy simulations (LES) simulations can take days of calculation on high-performance computer clusters. In contrast, the proposed farm power model is capable of predicting farm power for a 1-minute time step in a few milliseconds on PC. To give an example where this high prediction speed is crucial, is where the farm power is used in a reinforcement learning (RL) control setting where fast evaluations of possible controller actions are required. Even a prediction speed in the order of 1s would lead to impractical training times for the RL agent. We will opt for better wording than 'real-time power forecast' to position this use case.

***Referee comment***: - *Line 7: I would specify the order of magnitude of the "multiple time horizons" in the abstract.*

**Response of authors**: We are not sure whether we understand correctly what you want to emphasize with this in the abstract. In fact, as long as a weather forecast or estimate is available, any time horizon can be chosen. The look-ahead time horizon could also be very short, if for example a scanning lidar is used to measure the wind a few kilometers upstream of the wind farm. In this work, we were using weather forecast data for intra-day, one-day ahead and three-days ahead.

***Referee comment***: - *Line 8: what do you mean by "predict the wake losses"? Do you mean the wake losses percentage or more detailed wake losses info (like in a steady-state simulation)?*

**Response of authors**: We mean the power losses due to wake effects (expressed in MW or percentage of the fleet capacity). We do not mean the spatial distribution of the wakes across the farm. We will clarify this in our revised manuscript.

***Referee comment***: - *Line 70: I would give some examples of different commercial weather forecast services, their differences and their advantages.*

**Response of authors**: We are not sure whether we understand well your suggestion. We may add in our revised manuscript some names of weather forecast providers as example. However, we do not want to publish publicly a comparison of commercial services.

***Referee comment***: - *Lines 131 to 138: I find this paragraph important but not clear enough. What about a table with the different data sources, showing the differences in a direct way (time resolution, accuracy, etc.).*

**Response of authors**: Thank you for your suggestion. In fact, lines 131 to 138 serve only as an introduction. The different data sources are described more in detail in each of the subsequent subsections. We will follow your suggestion and add a summarizing table in the revision of our manuscript.

***Referee comment***: - *Line 154: "an additional data field is available that expresses", fix formulation.*

**Response of authors**: We will adapt the formulation.

*Referee comment: - Line 182: please finish your sentence by adding a point at the end (and eventually commas at the end of each enumeration, except the last one).*

**Response of authors**: We will add punctuation.

*Referee comment: - Line 219: this Section is about the methodology and not the actual data sources, it shouldn't be here.*

**Response of authors**: We will move this section.

*Referee comment: - Line 246: do you mean "8 filters" instead of "8 kernels"?*

**Response of authors**: We will write "8 convolution filters" to improve readability.

*Referee comment: - Line 299: this part is not clear, what is "an equivalent model for a single turbine"? Is it another deep neural network for a single turbine?*

**Response of authors**: It is indeed another deep neural network for a single turbine. With "equivalent" we wanted to express that the model has the same data sources and time steps like the model for the farm. We will revise this formulation.

*Referee comment: - Line 305: this part describes the turbine model, but it should come before the "farm internal wake loss". And the "farm internal wake loss" is not really an "ML model" as it is just a formula using the output of 2 ML models. For clarity, I wouldn't put it in the "ML models" sub-Section.*

**Response of authors**: As mentioned in a previous suggestion, we agree to re-arrange the order of these subsections to improve readability.

*Referee comment: - Line 332: it is interesting to adapt weather forecast service to a specific wind farm using an ML-model. But what is the ML-model used, and why give only 2 lines of explanations for this?*

**Response of authors**: As mentioned in a previous suggestion, we will describe how we have implemented this model in the revised manuscript.

*Referee comment: - Line 341: the computing hardware and software is not part of the methodology but about the results / simulations / experiments.*

**Response of authors**: We will shift this subsection to the "Results" section.

*Referee comment: - Line 344: I am not familiar with operations, but it seems to be a strong assessment. It is enough to declare that your model can be used by wind farm operators?*

**Response of authors**: We believe that you are referring to the sentence: "This shows that the proposed model can be run on hardware that is readily accessible to wind farm operators." As the model can be run on a standard notebook, we believe that farm operators could, in principle, run such a model as well. This in contrast to models that need high performance computer clusters, such as large-eddy simulations (LES). We will reformulate the statement to reflect this comparison with high-compute models.

*Referee comment: - Line 254: why cannot you give precise numbers for the quantities of turbines?*

**Response of authors**: Due to confidentiality reasons, we cannot disclose the wind farm used for our assessment, and thus we cannot provide the exact number of turbines. We do provide an approximate number to reflect the order of magnitude of wind farms for which our model is constructed.

*Referee comment: - Line 399: you introduce 2 new deep neural networks quite fast here. It should be introduced in the methodology, as 2 baselines. And you need to give more details about their architecture and how they differ from your model.*

**Response of authors**: Thank you for your suggestion. We will add a separate subsection "Baselines" in the section "Methodology".

---

## Author Response (AR2)

**Reply to the reviewers**

We would like to thank all reviewers for their thorough analysis of our manuscript and their constructive comments. We have carefully considered each comment and provide below our answers. We will revise our manuscript accordingly and believe that this revision will address all reviewers' concerns.

**Anonymous Referee #2**

Report #2

Accepted as is.

**Anonymous Referee #3**

**Report #1**

Thank you for all the corrections, I find the paper clearer, but I still have some comments.

**Referee comment: Line 11**

I am still uncomfortable with the "faster than real-time power forecasting" formulation. I propose: "a key advantage of the data-driven approach is the high prediction speed compared to physics-based methods, such that it can be employed for applications where quasi-immediate power forecasting on different time scales is required".

**Response of authors**: Thank you for your suggestion. In the revised manuscript we will reformulate "faster than real-time power forecasting" in the abstract as follows: "enabling its use in applications that require forecasting multiple scenarios in real-time."

Further the document we specify the prediction speed more precisely ("a few milliseconds on PC") and we refer to the "reinforcement control setting where fast evaluations of many possible controller actions are required".

**Referee comment: Line 235**

You introduce 2.2.3 before 2.2.2, please correct the order.

Response of authors: We will correct the order in the revised document.

**Referee comment: Line 350**

I understand why you put the farm internal wake loss into sub-Section 2.2. But I am still uncomfortable by defining the farm internal wake loss as a "machine learning model". It uses the outputs of 2 different wake models, but it is not itself a ML model. You do not have another "ML model" specific to the wake losses, you do not have a loss function including the wake losses, etc. So I would put the farm internal wake loss outside the 2.2 ML models sub=Section. **Response of authors**: We agree and will put the farm internal wake loss outside of the sub-section with all ML models.

**Referee comment: Line 371**

I would put 2.2.6 at the beginning of the sub-Section 2.2. So you would introduce 2.2 directly by presenting the full model. You can also discuss the way you compute the farm internal wake loss. And only then, you detail each model individually, 2.2.1 wind farm power model, etc.

**Response of authors**: Thank you for your suggestion. In the revised manuscript, we will present the overall pipeline indeed before detailing every ML model separately. In line with your previous comment, we will put the farm internal wake loss outside of the ML-models sub-section.

**Referee comment: Line 390**

General comment: I feel that the results section is a little too big compared to the size of the paper. I would, if possible, reduce this section by focusing on the most important results and put less important results into an annex.

**Response of authors**: In the revised manuscript, we will shift the tables with the hyper-parameters of the ML models to an annex. While these details are valuable for reproducing our work, they are less important to the overall understanding and flow of the manuscript.